# Efficacy and safety of electrical acupoint stimulation for postoperative nausea and vomiting: A systematic review and meta-analysis

**Liyue Lu[1], Chenlong Xie[1], Xing Li[2‡], Yalan Zhou[2‡], Zhiyu Yin[2‡], Pan Wei[2‡], Hao Gao[2‡], Jian Wang[2]\*, Yue Yong[1,2]\*, Jiangang Song[1,2]\***

**1** Department of Anesthesiology & Research Institute of Acupuncture Anesthesia, Shuguang Hospital Affiliated with Shanghai University of Traditional Chinese Medicine, Shanghai, China, **2** Department of Anesthesiology, Shuguang Hospital Affiliated with Shanghai University of Traditional Chinese Medicine, Shanghai, China

☯ These authors contributed equally to this work.
‡ XL, YZ, ZY, PW and HG also contributed equally to this work.
\* songjg1993@shutcm.edu.cn (JS); yy_517@163.com (YY); wj_3096@shutcm.edu.cn (JW)

**Data Availability Statement:** All relevant data are within the manuscript and its Supporting Information files.

## Abstract

### Background

Postoperative nausea and vomiting are typical postsurgical complications. Drug therapy is only partially effective. The goal of our meta-analysis is to systematically evaluate the efficacy and safety of electrical acupoint stimulation for postoperative nausea and vomiting and to score the quality of evidence supporting this concept.

### Methods

PubMed, Embase, Cochrane Library, Web of Science, and ClinicalTrials.gov were searched from inception to March 19, 2020.

### Results

Twenty-six studies (2064 patients) were included. Compared with control treatment, electrical acupoint stimulation reduced the incidence of postoperative nausea and vomiting (RR 0.49, 95% CI 0.41 to 0.57, P < 0.001), postoperative nausea (RR 0.55, 95% CI 0.47 to 0.64, P < 0.001) and postoperative vomiting (RR 0.56, 95% CI 0.45 to 0.70, P < 0.001). Electrical acupoint stimulation also reduced the number of patients requiring antiemetic rescue (RR 0.60, 95% CI 0.43 to 0.85, P = 0.004). No differences in adverse events were observed. Subgroup analysis showed that both electroacupuncture (RR 0.58, 95% CI 0.46 to 0.74, P < 0.001) and transcutaneous electrical acupoint stimulation (RR 0.44, 95% CI 0.34 to 0.58, P < 0.001) had significant effects. Electrical acupoint stimulation was effective whether administered preoperatively (RR 0.40, 95% CI 0.27 to 0.60, P < 0.001), postoperatively (RR 0.59, 95% CI 0.46 to 0.76, P < 0.001), or perioperatively (RR 0.50, 95% CI 0.37 to 0.67, P < 0.001). The quality of evidence was moderate to low.

**Funding:** This work was supported by the National Natural Science Foundation of China [81774108, 81703898, 81973652].

**Competing interests:** The authors have declared that no competing interests exist.

## Conclusions

Electrical acupoint stimulation probably reduce the incidence of postoperative nausea and vomiting, postoperative nausea, postoperative vomiting, and reduce the number of patients requiring antiemetic rescue, with few adverse events.

## Introduction

Postoperative nausea and vomiting (PONV) are common postsurgical complications, and the incidence of PONV is approximately 20–30% [1], which increases up to 80% in high-risk patients without prophylactic antiemetic drugs [2]. Anesthetic factors, such as volatile anesthetics, opioids, type of surgery, and patient-related factors, are considered to have a key impact on the risk of PONV [3]. PONV not only reduces patient satisfaction after surgery but can also lead to serious postsurgical complications, such as electrolyte imbalance, aspiration of the gastric contents, increased intracranial pressure, aspiration pneumonia, suture dehiscence, and bleeding [4–11]. These factors ultimately prolong hospital stay and increase the cost of hospitalization [12].

PONV is triggered by several receptor systems, thus, preventing and treating PONV is a complex task [13, 14]. Antiemetic medications have widely been used to prevent PONV [15, 16]. They can be grouped into six different classes: 5-hydroxytryptamine 3 receptor antagonists, dopamine-2 receptor antagonists, neurokinin 1 receptor antagonists, corticosteroids, antihistamines, and anticholinergics [17–20]. Currently, most hospitals use antiemetic therapy involving a combination of these drugs [14]. Using dexamethasone in the beginning and 5-HT3 antagonist at the end based on Apfel Risk score is the best method to prevent PONV [14]. In addition, there are also pharmacological treatment modalities such as supplemental crystalloids [21], chewing gum [22], and ginger [13]. However, there is a complex and challenging aspect to pharmacological prophylaxis of PONV since it requires consideration of the individual patient's PONV risk as well as the pharmacokinetics, efficacy, adverse effects profile, cost-effectiveness, and availability of antiemetic agents [14]. There are non-pharmacologic approaches to PONV prevention as well. For example, acupuncture point stimulation [23, 24] and *M*orinda citrifolia Linn [25], etc. These nonpharmacological strategies offer attractive methods for managing PONV. In particular, acupoint stimulation shows promise for the management of PONV [16].

Some studies indicate that traditional Chinese medicine, especially acupoint stimulation, could be useful for prophylaxis and treatment of PONV [26, 27]. Electroacupuncture (EA) and transcutaneous electrical acupoint stimulation (TEAS) are commonly used interventions for acupoint stimulation [28, 29]. What EA and TEAS have in common is that they both apply electrical acupoint stimulation (EAS). A major difference between these interventions is that the former is invasive (involving needle insertion into the skin), whereas the latter is not. Clinically, both of these interventions are potential candidates for treating PONV [23, 30, 31]. EA and TEAS can be collectively called EAS, an effective and quantifiable modern version of manual acupuncture.

However, there is still insufficient evidence to demonstrate that electrical stimulation at acupuncture points has efficacy and safety for treating PONV. Hence, we systematically searched for published articles related to this treatment and conducted a systematic review of the evidence.

## Methods

Our review was registered with PROSPERO (www.crd.york.ac.uk/PROSPERO) with the registration number CRD42020181386. This study was conducted in accordance with Preferred

Reporting Items for Systematic Reviews and Meta-Analyses [32] and Assessing the methodological quality of systematic reviews (AMSTAR) guidelines [33].

## Literature search

A comprehensive literature search was conducted of the major databases PubMed, EMBASE, Cochrane Library, Web of Science, and ClinicalTrials.gov from inception to March 19, 2020 without language restrictions. We used the following words: ("electroacupuncture [Mesh]" OR "electric stimulation therapy [Mesh]" OR "transcutaneous electric nerve stimulation [Mesh]" OR " electrical acupuncture" OR "electro-acupuncture" OR "TENS" OR "TEAS" OR "electroanalgesia*" OR "electric* stimulat*") AND ("surgical" OR "preoperative care" OR "procedure" OR "surgery" OR "preoperative") AND ("nause*" OR "vomit*" OR "emesis" OR "emeses" OR "emet*" OR "queas*"). Other databases use their own subtitles for individual searches individually to determine all eligible studies (S1 Table).

The control treatment (no acupuncture + active device, no acupuncture + inactive device, gel electrodes + inactive device, or usual care) as the control group: 1. No acupuncture + active device means the needles were bent to lay flat against the skin, or no needles were applied. Insulated wires from the activated stimulator box with a normal current output were attached to the needles or the inside of the arm covers. 2. No acupuncture + inactive device means the needles were bent to lay flat against the skin, or no needles were applied. Insulated wires from the inactivated stimulator box with no current output were attached to the needles or the inside of the arm covers. 3. Gel electrodes + inactive device means gel electrodes were placed on the acupoints and attached to a stimulator box with no current output.

## Inclusion and exclusion criteria

The inclusion criteria were specified by the Population, Intervention, Control, Outcomes, and Study design (PICOS). The inclusion criteria were: 1. The review included emergency and elective surgery patients that underwent general anesthesia, regardless of age, sex, ethnicity, or surgery type. 2. The intervention measure in the treatment group was EAS (EA or TEAS). 3. the intervention measures in the control group were sham acupuncture, sham acupoint, sham electrical stimulation, or preoperative routine nursing. 4. Primary outcomes were the incidence of PONV, postoperative nausea (PON), or postoperative vomiting (POV). Secondary outcome measures were the numbers of patients requiring antiemetic rescue and adverse events. 5. The study was a randomized controlled trial (RCT) (blinded or non-blinded).

The exclusion criteria were as follows: 1. Patients undergoing cesarean section or abortion or those in the control group who received any electrical stimulation. 2. American Society of Anesthesiologists (ASA) $\geq$ III. 3. Literature with incorrect data or inaccessible data.

## Study selection and data collection

Two researchers independently used EndNote (version X8.1, Clarivate Analytics, Philadelphia, United States) reference management software for literature classification, preparation, and removal of duplicates. Then, we excluded nonrelevant studies that did not meet the inclusion criteria after reading the title and abstract of each article. The articles that could not be excluded based on the title and abstract were retrieved for a full-text screening. If necessary, we acquired additional information from the trial authors by email or telephone.

A standardized data extraction list in an Excel spreadsheet (Microsoft Corporation, Redmond, Washington, United States) was used to collect information from the included studies: author, year, country, study design, sample size, age, types of surgery, intervention measures, duration and outcome indicators. This process was performed by two researchers

independently and cross-checked after completion. When opinions differed, a third reviewer was consulted, and the case was reviewed for consensus.

## Outcomes

The primary outcomes were the incidence of PONV, PON, or POV. The secondary outcome measures were the numbers of patients requiring antiemetic rescue and adverse events.

## Risk of bias assessment

The risk of bias of the clinical trial was independently assessed by two reviewers using "The Cochrane Collaboration's tool" [34], and any disagreements were solved by discussion. Furthermore, publication bias was evaluated by funnel plot analyses if sufficient studies were included.

## The Grades of Recommendation Assessment Development and Evaluation

The quality of the evidence was assessed by the Grades of Recommendation Assessment Development and Evaluation (GRADE) system [35]. The GRADE system includes the risk of bias, inconsistency, indirectness, inaccuracy, and publication bias [36]. Bias risks included inappropriate randomization methods, allocation concealment, blindness, and excessive loss of follow-up data. The inconsistencies mainly involved different intervention or evaluation metrics. Indirectness covered direct and indirect comparisons of results between two groups. The inaccuracy was mainly judged by the width of the confidence intervals. Publication bias was due to unpublished studies (usually a negative result) by the investigator.

For each outcome, we initially awarded four points to each RCT and then downgraded the point total for defects regarding the 5 aspects. The quality of evidence was classified as A (high quality), B (medium quality), C (low quality), or D (very low quality) [37].

## Statistical analysis

Statistical analyses were carried out using RevMan 5.4 (RevMan, the Cochrane Collaboration, Oxford, United Kingdom) software. We examined 5 outcomes, each as dichotomous variables. The differences in categorical outcomes between the EAS and control groups were reported as the relative risk (RR) with a 95% confidence interval (CI). Significant heterogeneity was considered when $I^2 > 50\%$ or $P < 0.1$, and a random-effects model was used for analysis [38]. Otherwise, a fixed-effects model was used.

If $I^2 > 50\%$ in a group of studies, suggesting there was high heterogeneity, then we used sensitivity analysis or subgroup analysis to investigate the possible reasons from both clinical and methodological perspectives [39].

## Results

### Study selection

A total of 864 articles were retrieved from the database search. By screening the article titles, abstracts, or both and removing duplicates, 794 records were excluded. Seventy full-text articles were included for rescreening. We excluded 3 reports published as abstracts, 15 articles lacking relevant data, 11 articles showing improper patients, 7 studies showing improper control groups, 2 studies without descriptions of acupoints, and 6 articles that were protocols. Finally, we included 26 studies [40–65]. A flowchart of the literature screening process is outlined in Fig 1.

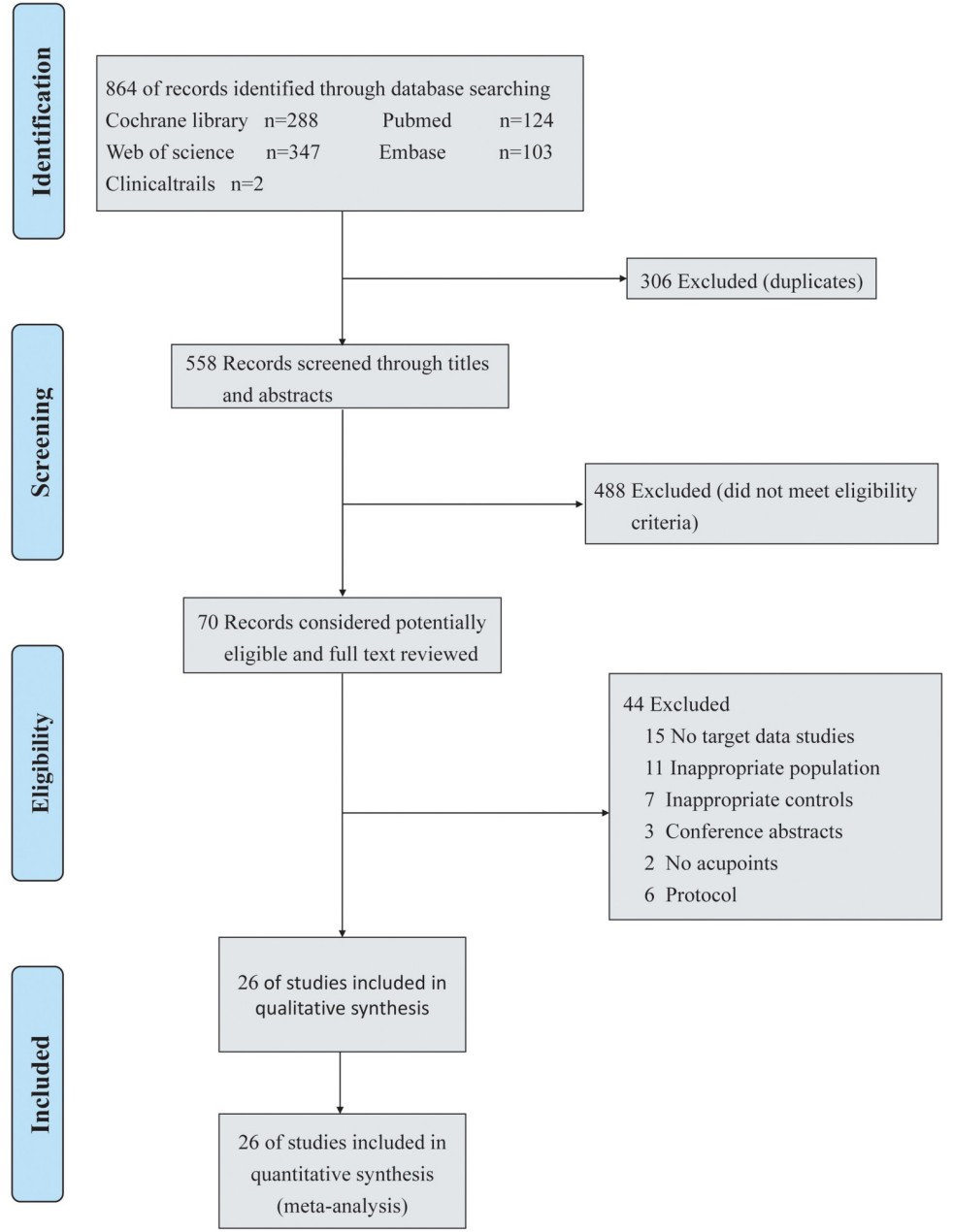

**Fig 1. Flow diagram of the literature search.**

## Description of the included studies

Twenty-six studies [40–65] met our inclusion criteria and included work performed in 8 countries (United States of America, The United Kingdom, China, India, Malaysia, Turkey, Iran, and Denmark) with a total of 2,064 patients. Among them, 1,051 patients received EAS, and 1,013 patients served as controls. The surgical type included laparoscopy, lithotripsy, tonsillectomy, thyroidectomy, gynecological, thoracotomy, sinusotomy, craniotomy, and plastic surgery. Among the included RCTs that used EAS as an intervention measure, 8 trials [46–50, 54, 56, 57] including 545 patients performed EA and 18 trials [40–45, 51–53, 55, 58–65] including

1519 patients performed TEAS. The control patients received perioperative routine nursing in 7 trials [41, 43, 46, 49, 50, 54, 56], TEAS with an inactive device in 16 trials [40, 42, 44, 45, 51–53, 55, 58–65], an active device without needles in 2 trials [47, 48], and an inactive device without needles in 1 trial [57]. The time to first treatment was before surgery in 8 studies [41, 51, 54, 59, 61–63, 65], after surgery in 7 studies [40, 42–44, 47, 53, 57], and during surgery in 11 studies [45, 46, 48–50, 52, 55, 56, 58, 60, 64]. The duration of first treatment lasted from 10 minutes to 24 hours. Nausea and vomiting were measured by the frequency of PONV, PON, and POV as the primary outcome and the use of rescue antiemetic and adverse events as secondary outcomes. The safety assessment involved the incidence of adverse events. However, 14 (53.85%) studies [46–50, 52, 55, 57, 59–64] lacked data on adverse events. Table 1 and S2 Table shows the characteristics of the included literature.

## Risk of bias within studies

All of the included trials mentioned randomization but 2 articles [43, 44] failed to report the method used to generate random sequences and were rated as unclear risk. For allocation concealment, 7 studies [48, 54, 57, 60–63] used a sealed envelope and 2 studies [52, 53] used central allocation and considered low risk, whereas the others did not mention allocation concealment and were rated as unclear risk. For blinding, 16 studies [40–47, 49–51, 55, 56, 58, 59, 64] provided insufficient information or used methods that could allow the patients to be aware of their assigned groups and were rated as unclear risk. For blinding of the outcome measurers, 5 studies [43, 53, 57, 64, 65] were rated as unclear risk, 3 [53, 57, 64] of which were not blinded. Considering the specificity of EAS treatment, blinding might not be done for operators. The remaining 2 studies [43, 65] did not describe blinding in their design. In addition, due to incomplete outcome data reporting, 6 studies [43, 44, 48, 50, 56, 64] did not describe the loss rate of the follow-up population, and 2 studies [42, 55] did not describe the reasons for the loss in the follow-up. These were, therefore, classified as unclear risk. One study [41] was judged as high risk due to later changes in random enrollment personnel. For selective reporting, 1 trial [47] did not describe adverse events of therapy, but when asked, the authors indicated that no obvious adverse effects were noted during the study. In terms of other biases, for the purpose of treatment, 3 studies [41, 42, 45] were rated as high risk. Among them, 1 article [42] was rated as high risk due to the differences between the groups. Another article [45] was rated as high risk due to the inconsistency of the data between figures and text. The last article [41] was rated as high risk due to the late supplementation of patients (see S1 Fig).

## Incidence of PONV

Sixteen of the studies [40, 41, 45–49, 52, 53, 55, 56, 58, 59, 61, 62, 65] included 1372 patients and measured and recorded the results of PONV after EAS. A fixed-effects model was used (P = 0.48; $I^2$ = 0%). The results indicated that participants who received EAS exhibited a significantly lower incidence of PONV than those in the control group (RR, 0.49; 95% CI, 0.41 to 0.57; P < 0.001) (Fig 2A). Sensitivity analysis of the outcomes showed that the heterogeneity was not significantly lower after any study was excluded (S3 Table).

According to the observation time, subgroup analysis showed that PONV within 24 h after surgery (RR, 0.51; 95% CI, 0.43 to 0.60; P < 0.001; $I^2$ = 0%) and at other times after surgery (RR, 0.40; 95% CI, 0.25 to 0.64; P = 0.001; $I^2$ = 30%) could be significantly reduced by EAS (Fig 2B).

The RCTs that recorded the incidence of PONV included several types of surgical procedures. To clarify the impact of the surgical procedure, we performed a subgroup analysis.

**Table 1. Characteristics of the included trials.**

| Author (year) | County | Study design | Surgery | Age (T/C) | Number of participants (T/C) | Intervention | | Time point | Target outcomes | Adverse events |
|---|---|---|---|---|---|---|---|---|---|---|
| | | | | | | Treatment group | Control group | | | |
| Yeoh et al, 2016 [40] | Malaysia | RCT | LS | 46.5 ±14.3/ 41.5 ±13.8 | 40/40 | TEAS | Inactive device | 24 h after surgery | 1, 4, 5 | No |
| Kabalak et al, 2005 [41] | Turkey | RCT | Tonsillectomy | 6.9±3.6/ 6.7±3.8 | 30/30 | TEAS | Usual care | 5 min before induction | 1, 5 | Erythema |
| Zárate et al, 2001 [42] | USA | RCT | LC | 42±16/39 ±14 | 110/56 | TEAS | Inactive device | 9 h after surgery | 2, 3, 5 | Erythema |
| Ye et al, 2008 [43] | China | RCT | Craniotomies | 44.5±15/ 35.8 ±20.2 | 20/20 | TEAS | Usual care | After surgery: 1 h per time, and 2 h intervals in 1 d | 2, 3, 5 | No |
| Chen et al, 1998 [44] | USA | RCT | AHS or myomectomy | 43±13/45 ±12 | 25/25 | TEAS | Inactive device | After surgery: 30 min per time, and 2–3 h intervals in 3 d | 2, 3, 4, 5 | Itching |
| Liu et al, 2008 [45] | China | RCT | LC | 42±18/40 ±19 | 48/48 | TEAS | Inactive device | 30~60 min before the induction till the end of surgery | 1, 2, 3, 4, 5 | No |
| An et al, 2014 [46] | China | RCT | SC | 40.7 ±12.1/ 39.1 ±10.9 | 41/40 | EA | Usual care | From induction to the end of surgery | 1 | ND |
| Rusy et al, 2002 [47] | USA | RCT | Tonsillectomy | 6.35 ±2.06/ 6.53 ±2.49 | 40/40 | EA | No acupuncture with active device | 20 min after surgery | 1, 2, 3, 4 | ND |
| Sahmeddini et al, 2010 [48] | Iran | RCT | Septoplasty | 27±11/29 ±10 | 45/45 | EA | No acupuncture with active device | 5 min each before surgery and till the end of surgery | 1 | ND |
| El-Rakshy et al, 2009 [49] | UK | RCT | LC or AHS | > 18 | 44/58 | EA | Usual care | During the period of induction | 1 | ND |
| Christensen et al, 1989 [50] | Denmark | RCT | AHS or ASO or tubal infertility | 41.5/41.5 | 10/10 | EA | Usual care | During the period of induction | 2 | ND |
| Zhang et al, 2014 [51] | China | RCT | ABS | 35.0±7.6/ 34.3±9.1 | 33/32 | TEAS | Inactive device | 30 min before induction | 2, 3, 5 | Pruritus |
| Tu et al, 2018 [52] | China | RCT | VATSL | 64.34 ±8.25/ 62.88 ±8.37 | 72/72 | TEAS | Inactive device | throughout the surgery | 1 | ND |
| Tu et al, 2019 [53] | China | RCT | Ureteroscopic lithotripsy | 64.32 ±10.21/ 62.14 ±11.34 | 60/60 | TEAS | Inactive device | 4,8,12 h postoperatively and three times on the next 2 d after surgery | 1, 5 | Itch |
| Li et al, 2017 [54] | China | RCT | GLS | 35.2±6.1/ 34.4±9.1 | 20/20 | EA | Usual care | 30 min before surgery | 2, 3, 5 | Pain, bruising |
| Gu et al, 2019 [55] | China | RCT | Laparoscopic radical gastrectomy | 57.59 ±7.32/ 56.67 ±6.23 | 58/59 | TEAS | Inactive device | 30 min before induction to 30 min after surgery: 30 min per time, and 3 times 1 day in 2d | 1 | ND |

(*Continued*)

**Table 1.** (Continued)

| Author (year) | County | Study design | Surgery | Age (T/C) | Number of participants (T/C) | Intervention | | Time point | Target outcomes | Adverse events |
|---|---|---|---|---|---|---|---|---|---|---|
| | | | | | | Treatment group | Control group | | | |
| Amir et al, 2007 [56] | India | RCT | Middle ear surgery | 17.95 ±8.25/ 21.1 ±7.48 | 20/20 | EA | Usual care | 20 min before induction till the end of surgery | 1, 2, 3, 4, 5 | Erythema |
| Chen et al, 2016 [57] | China | RCT | Lobectomy surgery | 55.80 ±15.75/ 56.93 ±14.61 | 46/46 | EA | No acupuncture with inactive device | after surgery: 30 min per time, and 2 times 1 day in 3d | 2, 3 | ND |
| Yang et al, 2015 [58] | China | RCT | GLS | 37/35 | 50/50 | TEAS | Inactive device | 30 min before surgery and lasting to leave PACU | 1, 2, 3, 4, 5 | Redness, swelling, itching |
| Yu et al, 2020 [59] | China | RCT | GLS | 48.5 ±16.2/ 45.9 ±17.5 | 30/30 | TEAS | Inactive device | 30 min before anesthesia | 1, 4 | ND |
| Liu et al, 2015 [60] | China | RCT | SC | 42.1 ±10.83/ 43.8 ±11.47 | 44/44 | TEAS | Inactive device | 30 min before anesthesia and lasting to leave PACU | 2, 3 | ND |
| Chen et al, 2015 [61] | China | RCT | Thyroidectomy | 41.9±9.9/ 41.4±9.8 | 29/30 | TEAS | Inactive device | 30 min before induction | 1 | ND |
| Chen Y et al, 2015 [62] | China | RCT | Thyroidectomy | 37.5±8.5/ 40.2±7.8 | 41/42 | TEAS | Inactive device | 30 min before induction | 1 | ND |
| Yao et al, 2015 [63] | China | RCT | GLS | 34.2±7.2/ 35.6±8.7 | 35/36 | TEAS | Inactive device | 30 min before induction | 2, 3 | ND |
| Zheng et al, 2008 [64] | China | RCT | Head and neck tumor surgery | 54.7 ±10.2/ 53.6±9.5 | 30/30 | TEAS | Inactive device | 30 min in the first 2–4 h, then once more every 3 h, 3 times in total. | 2, 3, 4 | ND |
| wang et al, 2014 [65] | China | RCT | Sinusotomy | 39.9/ 43.1 | 30/30 | TEAS | Inactive device | 30 min before anesthesia | 1, 4 | Pruritus |

Note: EA, electroacupuncture; TEAS, transcutaneous electrical acupoint stimulation; T, treated group (included EA and TEAS); C, control group; LC, laparoscopic cholecystectomy; LS, laparoscopic surgery; GLS, gynecological laparoscopic surgery; SC, supratentorial craniotomy; AHS, abdominal hysterectomy surgery; ABS, ambulatory breast surgery; ASO, hysterectomy, salpingo-oophorectomy; VATSL, video-assisted thoracic surgical lobectomy. * 1: postoperative nausea and vomiting; USA, United States of America; UK, The United Kingdom. 2: postoperative nausea; 3: postoperative vomiting; 4: number of patients requiring antiemetic rescue; 5: incidence of adverse effects. * ND = Not defined

Subgroup analysis showed that for laparoscopy (RR, 0.46; 95% CI, 0.36 to 0.60; P < 0.001; $I^2$ = 0%), thyroidectomy (RR, 0.40; 95% CI, 0.24 to 0.65; P = 0.0002; $I^2$ = 0%), and tonsillectomy (RR, 0.69; 95% CI, 0.53 to 0.90; P = 0.007; $I^2$ = 0%), EAS reduced the incidence of PONV (P < 0.001, S2A Fig).

In 25% (4/16) of the included studies, the control patients did not receive placebo targeting acupuncture points or electrical stimulation. Thus, we conducted a subgroup analysis. The results showed that with placebo (no acupuncture + active device, no acupuncture + inactive device, or gel electrodes + inactive device) (RR, 0.48; 95% CI, 0.40 to 0.57; P < 0.001; $I^2$ = 12%) or without placebo (usual care only) (RR, 0.53; 95% CI, 0.33 to 0.84; P = 0.007; $I^2$ = 0%), EAS reduced the incidence of PONV (P < 0.001, S2B Fig).

PONV within 24 h contained the maximum number of studies [40, 41, 45–47, 49, 56, 58, 59, 61, 62, 65]. We further used these studies to perform subgroup analysis based on different

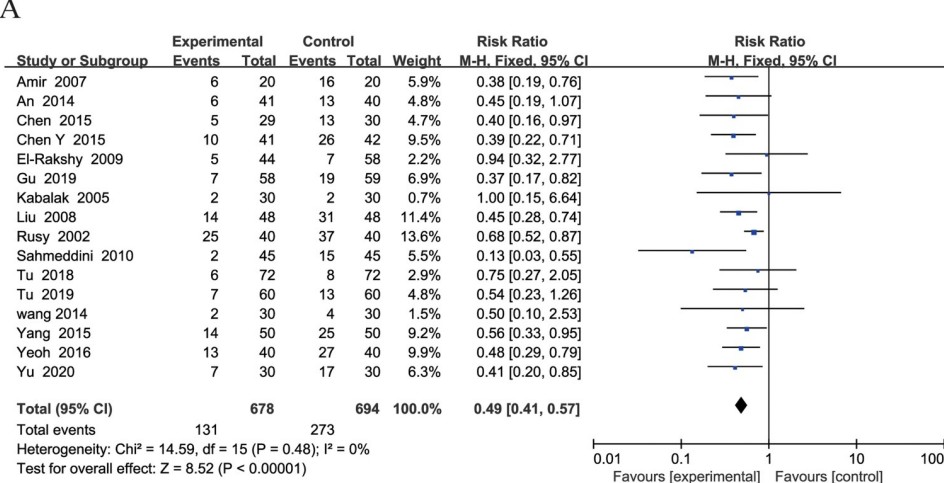

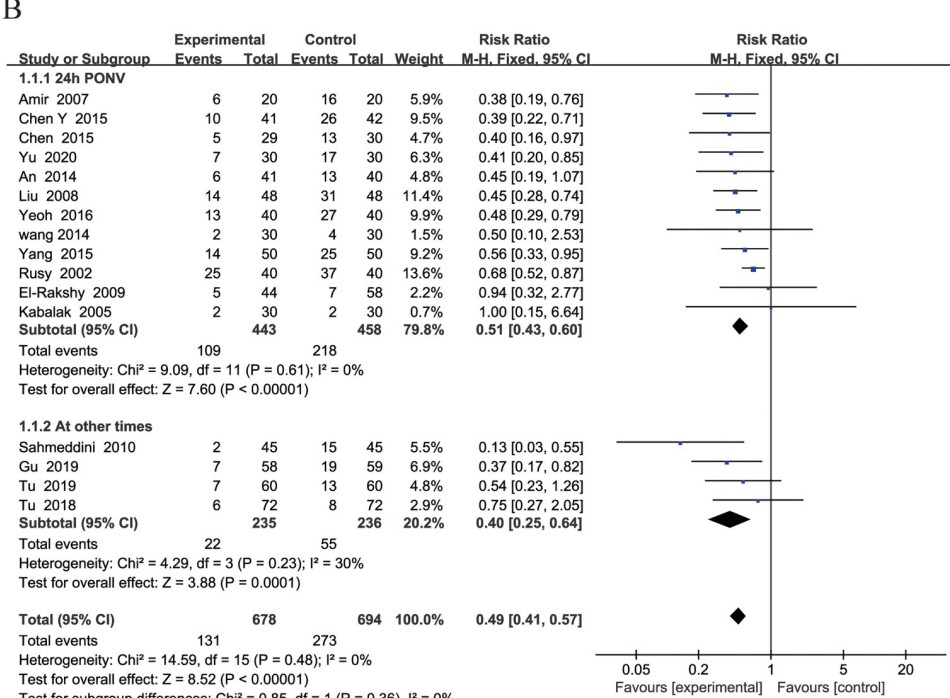

**Fig 2. Forest plots of the incidence of postoperative nausea and vomiting in EAS vs control.** A: Forest plots comparing the incidence of postoperative nausea and vomiting between EAS and Control group. B: Forest plot comparing the incidence of nausea and vomiting after EAS treatment within 24 hours after surgery and other times after surgery.

EAS therapies to determine whether EA or TEAS had the best treatment effect. A fixed-effects model was used ($I^2 = 0\%$). The results showed that both EA (RR, 0.58; 95% CI, 0.46 to 0.74; P < 0.001) and TEAS (RR, 0.45; 95% CI, 0.34 to 0.58; P < 0.001) had significant therapeutic effects (P < 0.001, Fig 3A). However, there was no significant difference between the two types of treatment (RR, 0.51; 95% CI, 0.43 to 0.60; P = 0.13).

According to the starting and ending times of the first intervention, the studies were divided into three subgroups: preoperative intervention, postoperative intervention, and

perioperative intervention groups. The results showed that preoperative interventions (RR, 0.43; 95% CI, 0.29 to 0.63; P < 0.001; $I^2$ = 0%), postoperative interventions (RR, 0.59; 95% CI, 0.46 to 0.76; P < 0.001; $I^2$ = 40%) and perioperative interventions (RR, 0.50; 95% CI, 0.37 to 0.67; P < 0.001; $I^2$ = 0%) all significantly reduced the incidence of PONV (Fig 3B). In this study, only 2 studies [40, 47] were in the postoperative group with a heterogeneity of 40%, and the other two groups had no heterogeneity ($I^2$ = 0%). It is possible that the intervention type and duration were different between the two postoperative studies.

### Incidence of PON

Thirteen studies [43–45, 47, 50, 51, 54, 56–58, 60, 63, 64] involving 842 participants reported the incidence of PON. The results were analyzed using a fixed-effects model (P = 0.03; $I^2$ = 46%). The results showed that the incidence of PON was significantly lower in the EAS group than in the control group (RR, 0.53; 95% CI, 0.45 to 0.62; P < 0.001) (Fig 4).

When the trial by zheng [64] was excluded, the heterogeneity decreased to 7%, with no change in the results. The remaining trials revealed that EAS could reduce PON compared to the control group. The sources of heterogeneity that remained may have been due to the different types of surgery and durations of the interventions (S3 Table).

### Incidence of POV

Twelve studies [43–45, 47, 51, 54, 56–58, 60, 63, 64] involving 822 participants reported the incidence of POV. The fixed-effects model analysis was conducted (P = 0.17; $I^2$ = 28%). The results demonstrated that the incidence of PON was significantly decreased in the EAS intervention group compared with the control group (RR, 0.56; 95% CI, 0.45 to 0.70; P < 0.001) (Fig 5).

Heterogeneity substantially disappeared when we excluded one study [47] (P = 0.54; $I^2$ = 0%), and the outcomes remained significant (RR, 0.49; 95% CI, 0.37 to 0.66; P < 0.001). It is possible that the intervention duration and type of operation were different (S3 Table).

### Number of patients needing antiemetic rescue

Nine studies [40, 41, 44, 45, 47, 56, 58, 59, 64] reported the number of patients who required antiemetic rescue. We analyzed the results using a random-effects mode (P = 0.04; $I^2$ = 51%). The results showed that the need for rescue antiemetics was significantly lower in the EAS group than in the control groups (RR, 0.60; 95% CI, 0.43 to 0.85; P = 0.004) (S3 Fig).

Most of the heterogeneity was caused by one study [47]. After exclusion, heterogeneity decreased to $I^2$ = 33%, which may have been due to the type of surgery, the method, and the duration of the intervention (S3 Table).

### Adverse events

Nine studies [41, 42, 44, 51, 53, 54, 56, 58] reported adverse reactions resulting from the treatment regimens. The heterogeneity test showed $I^2$ = 50%; P = 0.04, therefore, the fixed-effects model was used for analysis. There was no statistical significance between these two groups (RR, 0.94; 95% CI, 0.62 to 1.42; P = 0.78) (S4 Fig).

Sensitivity analysis of the outcomes showed that heterogeneity was not significantly lower after any study was excluded (S3 Table).

### Publication bias

No obvious release bias was found through the inverted funnel chart (S5 Fig).

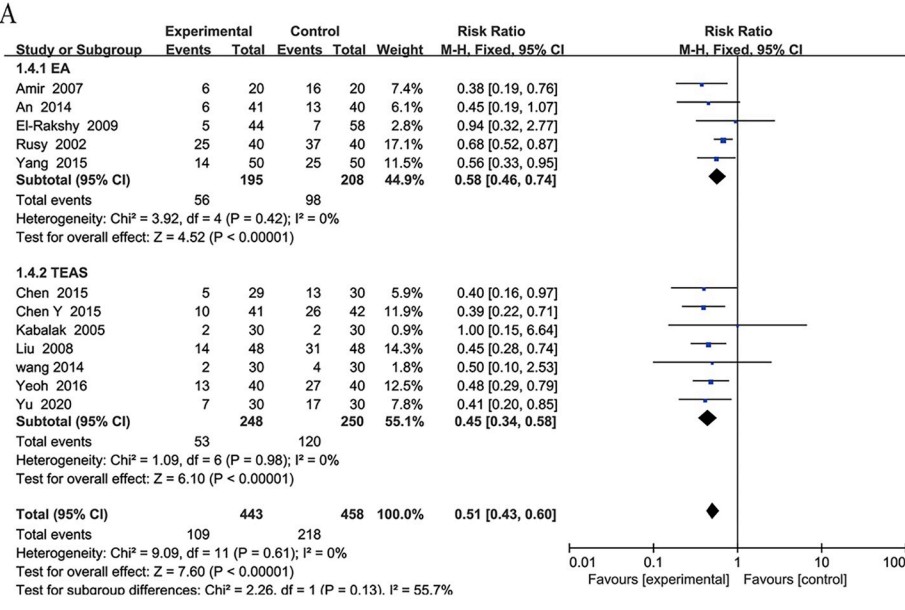

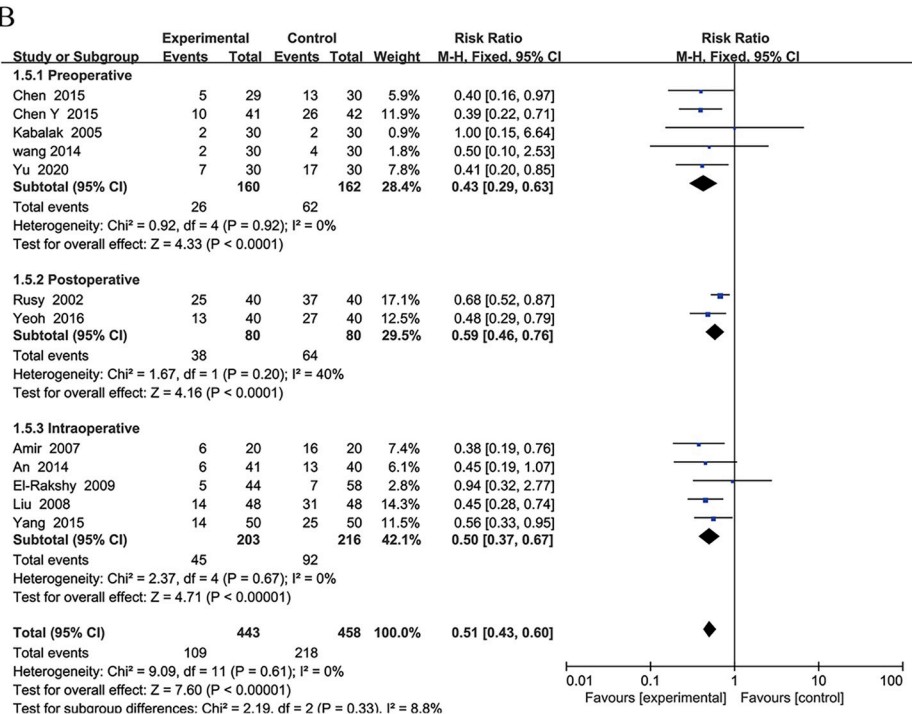

**Fig 3. Forest plots of the incidence of postoperative nausea and vomiting within 24 h. A:** Forest plots comparing the incidence of postoperative nausea and vomiting within 24 h between TEAS and EA. B: Forest plots comparing the incidence of postoperative nausea and vomiting within 24 h preoperatively, postoperatively, and perioperatively.

## Sensitivity analysis

S3 Table describes the details of the sensitivity analyses by excluding one study at a time. S4 Table shows the results of sensitivity analysis by changing the effects model for outcome analysis.

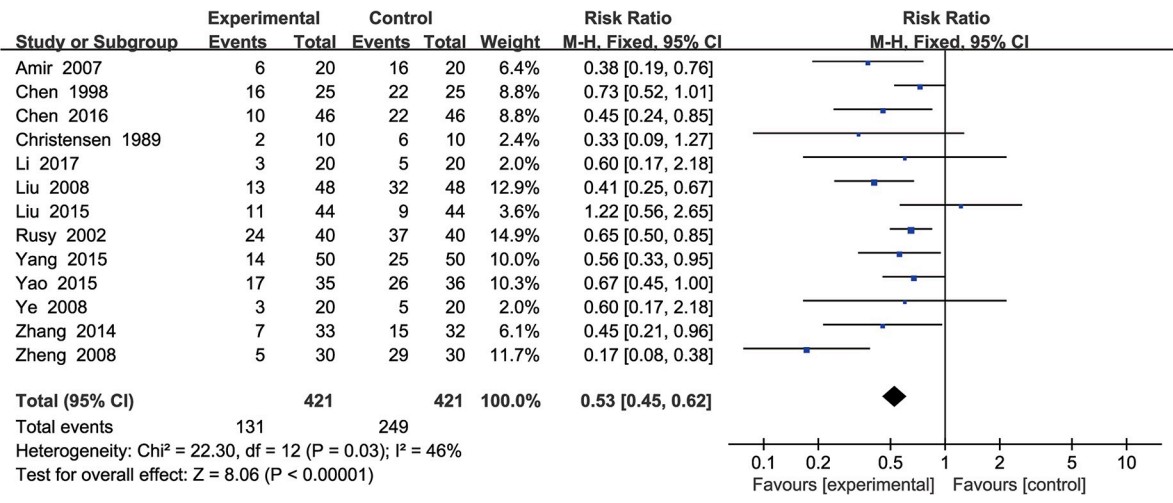

**Fig 4. Forest plots of the incidence of postoperative nausea in EAS vs control.**

## GRADE evaluation

Based on the principles of the GRADE evaluation, we evaluated the quality of the evidence provided via the PONV, 24 h PONV, PON, POV, rescue antiemetic, and adverse events assessments. Table 2 shows that, except for rescue antiemetic assessments, which were classified as low-quality, the others were evaluated as moderate in quality.

## Discussion

The results of this meta-analysis indicate that EAS significantly decreases the incidence of PONV, PON, and POV. Our results were consistent with the previous studies. A meta-analysis first published in 2004 was updated in 2015 and found that the effect of acupoint stimulation is comparable to antiemetics in preventing PONV [23]. But, this research only focused on evaluating the effect of PC6. Another meta-analysis indicated that additional effective meridian points in the treatment of PONV included BL10-11, BL18-26, SP4, SP6, ST34, ST36, ST44, and others [66], and thus PC6 may not be the only acupoint effective in treating PONV. We

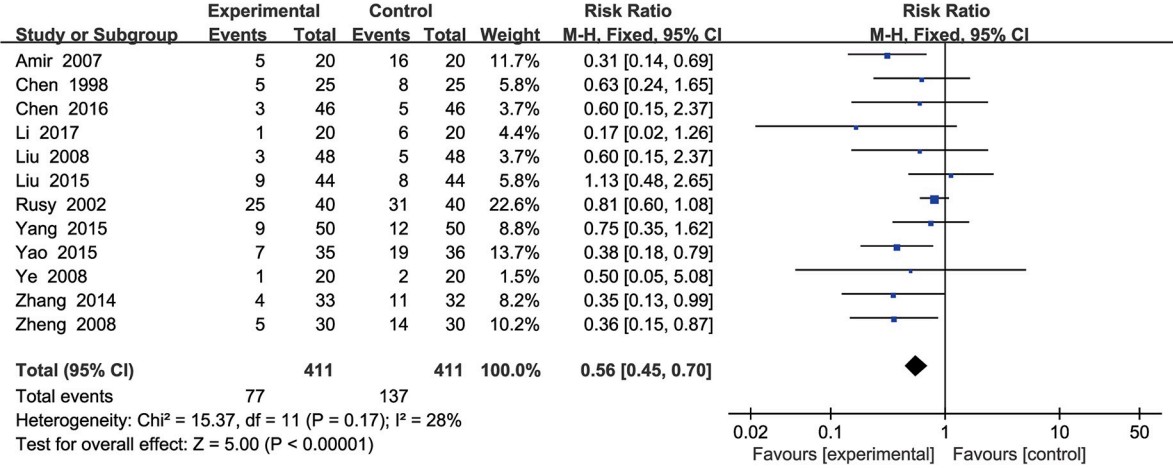

**Fig 5. Forest plots of the incidence of postoperative vomiting in EAS vs control.**

**Table 2. GRADE evaluation of evidence quality.**

| Outcomes Participants (studies) | Risk of bias | Inconsistency | Indirectness | Imprecision | Publication bias | Overall quality of evidence |
|---|---|---|---|---|---|---|
| PONV 1372(16 studies) | serious[1] | no serious inconsistency | no serious indirectness | no serious imprecision | no publication bias[2] | ⊕⊕⊕⊖ MODERATE[1,2] |
| 24h PONV 901(12 studies) | serious[3] | no serious inconsistency | no serious indirectness | no serious imprecision | no publication bias[2] | ⊕⊕⊕⊖ MODERATE[2,3] |
| PON 842(13 studies) | serious[4] | no serious inconsistency | no serious indirectness | no serious imprecision | no publication bias[2] | ⊕⊕⊕⊖ MODERATE[2,4] |
| POV 822(12 studies) | serious[5] | no serious inconsistency | no serious indirectness | no serious imprecision | no publication bias | ⊕⊕⊕⊖ MODERATE[5] |
| Rescue Antiemetic 616(9 studies) | serious[6] | serious[7] | no serious indirectness | no serious imprecision | no publication bias | ⊕⊕⊖⊖ LOW[6,7] |
| Adverse Events 700(9 studies) | serious[8] | no serious inconsistency | no serious indirectness | no serious imprecision | no publication bias | ⊕⊕⊕⊖ MODERATE[8] |

Note

[1] 11 studies were rated as unclear risk in allocation concealment; 2 studies were rated as hight-risk and 8 were rated as unclear in terms of other biases

[2] Funnel plot showed publication bias

[3] 10 studies were rated as unclear risk in allocation concealment; 2 studies were rated as hight-risk and 5 were rated as unclear in terms of other biases

[4] 2 studies were rated as unclear risk in random sequence generate; 8 studies were rated as unclear risk in allocation concealment; 4 studies applied a single-blind study design in blinding; 1 study was rated as hight-risk and 5 were rated as unclear in terms of other biases

[5] 2 studies were rated as unclear risk in random sequence generate; 7 studies were rated as unclear risk in allocation concealment; 4 studies applied a single-blind study design in blinding; 1 study was rated as hight-risk and 4 were rated as unclear in terms of other biases

[6] 1 study was rated as unclear risk in random sequence generate; 5 studies were rated as unclear risk in allocation concealment; 2 studies applied a single-blind study design in blinding; 2 study were rated as hight-risk and 2 were rated as unclear in terms of other biases

[7] I square = 51%

[8] 1 study was rated as unclear risk in random sequence generate; all studies were rated as unclear risk in allocation concealment; 2 studies applied a single-blind study design in blinding; 2 studies were rated as hight-risk and 3 were rated as unclear in terms of other biases.

included studies looking at many acupuncture points, which are more comprehensive and practical for studying the efficacy of EAS. The acupoints should be selected for each patient according to their presenting symptoms and characteristics.

Subgroup analysis showed that EAS, whether applied as a preoperative intervention, postoperative intervention, or perioperative intervention, had a significant effect. To the best of our knowledge, no meta-analysis has been conducted on this topic. However, one RCT showed that EA on PC6 is effective in the prevention of PONV, and pre-operative acupuncture is more effective than post-operative acupuncture. There are two possible reasons for the disparity in results. Firstly, meta-analysis can only show that the three types of treatment timing are effective because they were not directly compared in the included studies by us. Secondly, the previous meta-analysis indicated that the antiemetic effect of acupuncture require treatment of awake rather than anesthetized patients [67]. Third, during surgery, the needle cannot be well protected, and the overall effect may be minimized. For safety reasons, it is ideal to provide acupuncture to conscious patients before surgery because most of the acupuncture points are located around the nerves, and deep punctures on unconscious patients may damage nerves.

Another subgroup analysis demonstrated that both EA and TEAS had good therapeutic effects compared with the control group. As far as we know, no similar study comparing the efficacy of EA and TEAS for PONV. However, a meta-analysis indicated that acupuncture therapy can reduce the risk of PONV in abdominal operation. Another meta-analysis suggested that TEAS was effective in preventing PONV [68]. Those results were consistent with

the results obtained in our study. The addition of EAS treatment can reduce the workload of doctors, allowing doctors to treat multiple patients at once, and mass production also makes the parameters used in treatment more accurate. EAS is divided into EA and TEAS. EA combines a pulsating electrical current with acupuncture to enhance acupoint stimulation, which is a more effective method for administering acupuncture [28]. The advantages of EA are that it preserves the therapeutic effect of traditional acupuncture based on increasing EAS and combines electro with physical stimulation generated by acupuncture at acupuncture points. However, these advantages also mean that this process of acupuncture requires professional acupuncturists to only use invasive needles. In addition, some patients who are sensitive to needles may experience syncope during treatment and may have complications, such as subcutaneous bleeding, persistent acid bloating, and a burning sensation at the acupuncture site. TEAS has advantages: there is no intrusive behavior and patients' acceptance is higher. TEAS also has disadvantages: only electrical stimulation can be applied and there is a lack of physical stimulation of needle penetration into the skin. Additionally, TEAS may cause minor allergies, such as skin flushing. Compared with EA, TEAS treatment is safer, more effective, and worthy of clinical promotion.

We have found that for laparoscopy, thyroidectomy, and tonsillectomy, EAS reduced the incidence of PONV. Similar findings have also been reported. Studies on the preventive and therapeutic effects of acupuncture on PONV for specific types of surgeries, and the findings all indicate that acupuncture is an effective and safe therapy for PONV [69, 70].

In our meta-analysis, we included more patients and RCTs. In addition to the inclusion of TEAS, RCTs evaluating the effectiveness of EA in the treatment of PONV were also added. Thus, our study provides a more comprehensive analysis than the 2020 TEAS meta-analysis [68]. Furthermore, we used the Quality Evaluation Tool approach to rate the quality of evidence to provide more reliable conclusions. In addition, we conducted a subgroup analysis to compare the pros and cons of EA and TEAS and distinguish the role of EAS in different types of procedures.

The quality of the RCTs included in this review was assessed by the GRADE system [35]. GRADE clearly differentiates the quality of each RCT from the overall strength of the evidence, making the accuracy of the analysis results clearer [71, 72]. Grading the strength of the evidence using the GRADE approach is becoming an important, recommended step in a comprehensive evidence evaluation and could increase the transparency of the clinical decision-making processes, especially when the quality of the evidence is poor or unclear [71, 73]. AMSTAR is a measurement tool created to assess the methodological quality of systematic reviews [74]. It is a reliable, valid, and critical assessment tool developed by AMSTAR in 2017 [75]. According to the GRADE criteria and AMSTAR 2 system, the quality of evidence was moderate for all outcomes, except the rescue antiemetic requirement (low-quality evidence). Therefore, the results lower our confidence in clinical decisions. Fortunately, the methodological quality was high. Incorporating all the evaluation results reported in the study, some studies may have a certain degree of bias. However, no obvious release bias was found through the inverted funnel chart.

Moreover, regarding adverse events, there was no significant difference between the EAS group and the control group. This result indicated that EAS is not related to any serious adverse events, although some patients may present minor hemorrhage or pain at the insertion site of the needle, or redness and itching may appear on the skin where the surface electrode was applied. The safety data available were limited, as 14 (53.85%) studies did not report any side effects of EAS.

Apfel and his colleagues developed a data-based assessment tool to predict the risk of PONV [76]. The tool assigns one point for each known risk factor (gender, smoking status,

history of PONV or motion sickness, and use of opioids for postoperative pain) [77]. They found that the presence of 1, 2, 3, and 4 of these risk factors increased the incidence of PONV by 21%, 39%, 61%, and 79%, respectively [76]. The tool classifies patients with 0–1, 2, or 3-plus risk factors into "low," "medium," and "high" risk categories, respectively. In 2020, the Fourth Consensus Guidelines for the Management of PONV concluded that "multimodal PONV prophylaxis in patients with 1 or 2 risk factors, in an attempt to reduce risk of inadequate prophylaxis" [14]. Two antiemetics are recommended for PONV prophylaxis in patients at medium risk and 3–4 antiemetics in patients at high risk [14]. In order to evaluate whether EAS is useful for patients with "low," "medium," or "high" risk of PONV, RCTs in which participants were stratified by Apfel score should be done.However, only one article mentioned the Apfel scoring system during patient inclusion among the included studies. It specifies the inclusion of patients with an Apfel score ≥ 2 including patients with a medium-high risk for PONV [40]. While none of the other 25 studies had that inclusion criterion. Thus, we could not conduct a subgroup analysis based on Apfel scores. We agree that the Apfel scoring system is very important. If these RCTs included Apfel scores in their inclusion criteria or stratified randomized groups of patients based on the Apfel scoring system, higher-quality results would have been obtained.

Opioids have been identified as an independent risk factor for the development of PONV [77]. The mechanism of action of EAS may involve the regulation of endogenous opioid release and other neurotransmitters [78]. Evidence has shown that EA treatment might lead to better analgesia and reduce opioid use [79]. This is a possible mechanism of how EAS reduces PONV.

We excluded patients who underwent cesarean section or abortion, because the abortion patients may have had pregnancy vomiting. Similarly, cesarean patients might be administered oxytocin. Oxytocin causes emesis. We only included patients who underwent general anesthesia, whereas most cesarean patients do not use this type of anesthesia.

Our review has several limitations. First, most of the included studies had small sample sizes, and the event rates of several outcomes were low. This limitation may lead to imprecise evidence. Additionally, using GRADE, we judged the quality of the evidence in the review to be only moderate or low. The GRADE evidence quality of the pooled results lowers our confidence in the utility of the evidence to guide clinical decisions. Third, some design flaws are obvious. For example, there are many types of surgery included in the literature. The results of the analysis of all surgical studies are more likely to lead to bias. And some results that were significant for the evaluation of PONV were not included (the frequency of PONV). Cause only one of the 26 studies included in our meta-analysis recorded the frequency of PONV. Fourth, no clear comparison with already recommended pharmacological treatment group. Fifth, no obvious risk assessment or use of the scoring system to stratify the included studies. In addition, a certain degree of heterogeneity was observed in this meta-analysis. We tried to reduce heterogeneity through subgroup and sensitivity analyses, but it has not yet been completely resolved.

## Conclusion

Our analysis showed that EAS can decrease the incidence of PONV, PON, and POV, as well as the number of patients requiring antiemetic rescue, and does not increase adverse events in patients undergoing elective surgery under general anesthesia. According to the GRADE criteria, the quality of evidence was moderate for all outcomes, except the rescue antiemetic requirement (low-quality evidence). These findings suggest that EAS may be considered an effective and safe treatment for PONV and that the EAS approach may be promising to

promote the recovery of patients after surgery. The reliability of these results for PONV needs to be further explored, and the quantity and quality of the included studies need to be improved.

## Supporting information

**S1 Checklist. PRISMA 2009 checklist.**
(DOC)

**S1 Fig. Risk of bias summary: Review authors' judgments of each included trial.**
(TIF)

**S2 Fig. Subgroup analyses about postoperative nausea and vomiting.** A: Forest plots of types of surgical procedures. B: Forest plot of EAS vs. placebo.
(TIF)

**S3 Fig. Forest plots of the numbers of patients needing antiemetic rescue in EAS vs control.**
(TIF)

**S4 Fig. Forest plots of the adverse events in EAS vs control.**
(TIF)

**S5 Fig. Funnel plot for the assessment of publication bias for postoperative nausea and vomiting.** A: Funnel plot for postoperative nausea and vomiting. B: Funnel plot for postoperative nausea. C: Funnel plot for postoperative vomiting.
(TIF)

**S1 Table. Search strategy.**
(DOCX)

**S2 Table. Details of the included trials for treatments methods.**
(DOCX)

**S3 Table. Sensitivity analyses by excluding one study at a time.**
(DOCX)

**S4 Table. Sensitivity analysis by changing the effects model for outcome analysis.**
(DOCX)

**S5 Table. AMSTAR 2: A critical appraisal tool for systematic reviews.**
(DOCX)

**S1 Raw data.**
(RAR)

## Author Contributions

**Conceptualization:** Yue Yong, Jiangang Song.

**Data curation:** Liyue Lu, Chenlong Xie, Xing Li, Yalan Zhou, Zhiyu Yin, Pan Wei, Jian Wang, Yue Yong.

**Formal analysis:** Liyue Lu, Chenlong Xie, Zhiyu Yin, Pan Wei.

**Funding acquisition:** Zhiyu Yin, Jiangang Song.

**Methodology:** Liyue Lu.

**Project administration:** Liyue Lu.

**Software:** Chenlong Xie.

**Validation:** Jiangang Song.

**Writing – original draft:** Liyue Lu, Chenlong Xie.

**Writing – review & editing:** Liyue Lu, Chenlong Xie, Xing Li, Yalan Zhou, Zhiyu Yin, Pan Wei, Hao Gao, Jian Wang, Jiangang Song.

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
