## [Decision Letter · Decision Letter 0]

9 Feb 2022

PONE-D-21-05426

Efficacy and safety of electrical acupoint stimulation for postoperative nausea and vomiting: A systematic review and meta-analysis

PLOS ONE

Dear Dr. Song,

Thank you for submitting your manuscript to PLOS ONE. After careful consideration, we feel that it has merit but does not fully meet PLOS ONE’s publication criteria as it currently stands. Therefore, we invite you to submit a revised version of the manuscript that addresses the points raised during the review process.

The manuscript has been evaluated by three reviewers, and their comments are available below.

The reviewers have raised a number of concerns that need attention. They request additional information on methodological aspects of the study (such as the inclusion of information on the sample size and response rate), revisions to the statistical analyses and they question the internal and external validity of the results reported.

Could you please revise the manuscript to carefully address the concerns raised?

We look forward to receiving your revised manuscript.

Kind regards,

Elisa Panada

Associate Editor

PLOS ONE

Journal Requirements:

Reviewers' comments:

Reviewer's Responses to Questions

**Comments to the Author**

1. Is the manuscript technically sound, and do the data support the conclusions?

Reviewer #1: Partly

Reviewer #2: Yes

Reviewer #3: Partly

2. Has the statistical analysis been performed appropriately and rigorously? 

Reviewer #1: I Don't Know

Reviewer #2: Yes

Reviewer #3: I Don't Know

3. Have the authors made all data underlying the findings in their manuscript fully available?

Reviewer #1: Yes

Reviewer #2: Yes

Reviewer #3: Yes

4. Is the manuscript presented in an intelligible fashion and written in standard English?

Reviewer #1: Yes

Reviewer #2: Yes

Reviewer #3: No

5. Review Comments to the Author

Reviewer #1: PONE-D-21-05426

Efficacy and safety of electrical acupoint stimulation for postoperative nausea and

vomiting: A systematic review and meta-analysis

Thank you for the opportunity to review this submission to the journal. I have a small nuber of comments.

The submission is similar to the meta-analysis of Chen (Chen J, Tu Q, Miao S, Zhou Z, Hu S. Transcutaneous electrical acupoint stimulation for preventing postoperative nausea and vomiting after general anesthesia: a meta-analysis of randomized controlled trials. International Journal of Surgery. 2020 Jan 1;73:57-64.) but has a more up to date search (up to March 2020 vs July 2019) and as expected has more publications under analysis. I would have expected all of the studies included by Chen to be included in this submission but they are not – approximately half are included. What is the difference in the inclusion/exclusion criteria.

After a brief search of the literature there were a number of publications that seem eligible for analysis such as those below. The second was published online in February 2020 and seems eligible.

H. Wang, Y. Xie, Q. Zhang, N. Xu, H. Zhong, H. Dong, L. Liu, T. Jiang, Q. Wang, L. Xiong, Transcutaneous electric acupoint stimulation reduces intra-operative remifentanil consumption and alleviates postoperative side-effects in patients undergoing sinusotomy: a prospective, randomized, placebo-controlled trial, BJA: British Journal of Anaesthesia, Volume 112, Issue 6, June 2014, Pages 1075–1082, https://doi.org/10.1093/bja/aeu001

Chen J, Zhang Y, Li X, Wan Y, Ji X, Wang W, Kang X, Yan W, Fan Z. Efficacy of transcutaneous electrical acupoint stimulation combined with general anesthesia for sedation and postoperative analgesia in minimally invasive lung cancer surgery: A randomized, double‐blind, placebo‐controlled trial. Thoracic cancer. 2020 Apr;11(4):928-34.

The findings are similar to that of Chen in terms of effect on PONV, nausea, vomiting and anti-emetic use. I would be interested to know what the authors suggestion is as to the proposed mechanism of effect. Is it simply an indirect effect through better analgesia and downstream opioid sparing which is emetogenic itself. Is it possible to deduce this from the data by comparing studies in which opioid consumption was measured.

Lastly is the anti-emetic effect more or less pronounced if anti-emetics are administered prophylactically. This would require an assessment of the anti-emetic interventions in the controls. I am not clear from the description of studies how the control groups were managed in terms of prophylaxis against PONV. It should be stated whether PONV prophylaxis was administered in the controls. Which studies were head to head RCTs of electroacupuncture vs anti-emetics and what was the meta-analytic result of these?

Figures 2 and 3 are very blurred and hard to read in the main body of the document.

Reviewer #2: The paper entitled Efficacy and safety of electrical acupoint stimulation for postoperative nausea and vomiting: A systematic review and meta-analysis is well designed and well performed meta-analysis. The methods used here are appropriate, the manuscript is clear, well-organized. Only minor remarks are listed below:

- avoid using abbreviations in the abstract (TEAS)

- lines 74 and 77: repetition

-line 80: why not sufficiently comprehensive? Explain.

- line 84: meanwhile is not the appropriate word.

- line 172: one category (3 articles) is missing

- line 183: 7 or 8 trials?

- lines 233 AND 237: figure numbering to be corrected

Reviewer #3: Comments to the authors:

1. English language still needs some refinement and editing.

2. Abbreviation used in the manuscript should have full forms at the time of first mentioning.

3. Some details should be discussed about assessment and severity of PONV using Apfel scoring system.

4. Not only the therapeutic measures, also commonly adopted preventive measures should also me mentioned.

5. Many comments made about EAS are not supported by reference, which should be.

6. Role of EAS outside China is questionable and thus not widely used, as claimed here, even now.

7. Uniqueness of your meta-analysis compared to previous ones should be mentioned in the result and discussion sections, not in the introduction section.

8. Did you use any filter during your literature search?

9. What do you mean by “sham” treatment in the control group?

10. Why were cesarean section and abortion excluded?

11. Regarding the analysis of incidence of PONV, what was used in the control group? Also, you have mentioned incidence with EAS than presenting data separately on EA and TEAS.

12. Did you do any subgroup analysis here between different types of surgery or not? Also was the anesthesia technique used standard in all the studies?

13. What do you mean by “EAS plus other treatments”? What are the other components of the other treatments? From the data I can see the additional other treatment did not decrease the incidence, rather increased the incidence of PONV a bit compared to only EAS. Why so?

14. In remaining 20% studies where you have mentioned that they received some treatment than placebo in the control arm, what are those interventions used?

15. Apart from the incidence, did you do any analysis of frequency of PONV or not?

16. Regarding timing of intervention used, not finding any difference between pre and postoperative EAS means, it does not have any better preemptive effect. Am I right? How do you explain this?

17. Targeted and comparative literature review is missing.

Strengths:

• Very relevant topic chosen for meta-analysis.

• Extensive literature search used.

• Extensive statistical analysis and subgroup analysis and multiple outcomes measured.

Weakness:

• English language used is very lucid, has several grammatical mistakes and lacked scientific tone.

• Some design flaws are quite obvious.

• No clarity on control or sham treatment group.

• No clear comparison with already recommended pharmacological treatment group.

• No obvious risk assessment or use of scoring system.

• Several repetitions of statistical methods in the methodology and result section. Especially the result section is confusing.

• The literature review is incomplete without targeted comparative analysis.

Verdict: Good intent of analyzing the efficacy of non-pharmacological measure in very relevant PONV which is still an intriguing issue despite abundance of antiemetics. But the manuscript still needs several modifications, refinement before I can accept it.

6. PLOS authors have the option to publish the peer review history of their article (what does this mean?). If published, this will include your full peer review and any attached files.

Reviewer #1: No

Reviewer #2: No

Reviewer #3: **Yes: **Pradipta Bhakta

---

## [Author Response · Author response to Decision Letter 0]

7 Apr 2022

General comments:

We thank the editor and reviewers for their helpful suggestions and we have addressed all the points raised in our revised manuscript, and the revisions are outlined below. In addition to addressing these comments, we have also carefully reviewed the manuscript throughout to ensure clarity and reduce redundancy and repetition, and comply with Nature Communications style and formatting. 

Editor and Reviewer comments:

Reviewer #1: Efficacy and safety of electrical acupoint stimulation for postoperative nausea and vomiting: A systematic review and meta-analysis

Thank you for the opportunity to review this submission to the journal. I have a small number of comments.

1. The submission is similar to the meta-analysis of Chen (Chen J, Tu Q, Miao S, Zhou Z, Hu S. Transcutaneous electrical acupoint stimulation for preventing postoperative nausea and vomiting after general anesthesia: a meta-analysis of randomized controlled trials. International Journal of Surgery. 2020 Jan 1; 73: 57-64.) but has a more up to date search (up to March 2020 vs July 2019) and as expected has more publications under analysis. I would have expected all of the studies included by Chen to be included in this submission but they are not – approximately half are included. What is the difference in the inclusion/exclusion criteria.

Reply：We appreciate your professional question. Due to the influence of anesthetic factors, type of surgery, and patient-related factors on PONV [1], we added 2 exclusion criteria compared to Chen's study: (1) Exclusion of patients with ASA ≥ III; (2) Excluding patients in the control group who received any electrical stimulation. That is why only about half of the studies included by Chen were included in our study. We’re sorry that the inaccurate description in the manuscript may lead to misunderstanding. We have revised the inclusion and exclusion criteria in the method accordingly (line 99 to 110). The reasons for excluding these articles are listed in the following table:

Thank you again for your helpful question.

[1] Apfel CC, Roewer N. Risk assessment of postoperative nausea and vomiting. Int Anesthesiol Clin. 2003;41(4):13-32. Epub 2003/10/24. doi: 10.1097/00004311-200341040-00004.

2. After a brief search of the literature there were a number of publications that seem eligible for analysis such as those below. The second was published online in February 2020 and seems eligible.

H. Wang, Y. Xie, Q. Zhang, N. Xu, H. Zhong, H. Dong, L. Liu, T. Jiang, Q. Wang, L. Xiong, Transcutaneous electric acupoint stimulation reduces intra-operative remifentanil consumption and alleviates postoperative side-effects in patients undergoing sinusotomy: a prospective, randomized, placebo-controlled trial, BJA: British Journal of Anaesthesia, Volume 112, Issue 6, June 2014, Pages 1075–1082, https://doi.org/10.1093/bja/aeu001

Chen J, Zhang Y, Li X, Wan Y, Ji X, Wang W, Kang X, Yan W, Fan Z. Efficacy of transcutaneous electrical acupoint stimulation combined with general anesthesia for sedation and postoperative analgesia in minimally invasive lung cancer surgery: A randomized, double‐blind, placebo‐controlled trial. Thoracic cancer. 2020 Apr;11(4):928-34.

Replay: Thank you very much for this comment. Regarding the first literature (H. Wang, Y. Xie et al, 2014), we are very sorry that we excluded it by mistake. After rechecking and finding this to be an oversight on our part, we have revised all results and figures in the full paper. We are so sorry for our mistake.

Regarding the second study (Chen J, Zhang Y et al, 2020), we excluded it because patients in the sham-TEAS group received 4 mA stimulation. Our exclusion criteria included the control group receiving electrical stimulation.

Again, we thank the reviewer for pointing out this literature omission.

3. The findings are similar to that of Chen in terms of effect on PONV, nausea, vomiting and anti-emetic use. I would be interested to know what the authors suggestion is as to the proposed mechanism of effect. Is it simply an indirect effect through better analgesia and downstream opioid sparing which is emetogenic itself. Is it possible to deduce this from the data by comparing studies in which opioid consumption was measured.

Replay: Thanks for your suggestion. In deed, opioids have been long identified as an independent risk factor for the development of PONV [2]. And evidences showed that EA treatment might lead to better analgesia and reducing opioid use. This is a possible mechanism of how EAS reduced PONV. However, it is hard to drop this conclusion because the amount of medication used was not explicitly given in a large part of the included literature. So we are sorry that can only add some discussion about the possible mechanism (line 405 to 409).

[2] Apfel CC, Laara E, Koivuranta M, Greim CA, Roewer N. A simplified risk score for predicting postoperative nausea and vomiting: conclusions from cross-validations between two centers. Anesthesiology. 1999;91(3):693–700. doi: 10.1097/00000542-199909000-00022.

4. Lastly is the anti-emetic effect more or less pronounced if anti-emetics are administered prophylactically. This would require an assessment of the anti-emetic interventions in the controls. I am not clear from the description of studies how the control groups were managed in terms of prophylaxis against PONV. It should be stated whether PONV prophylaxis was administered in the controls. Which studies were head to head RCTs of electroacupuncture vs anti-emetics and what was the meta-analytic result of these?

Replay: Thanks for your question. In the included studies, the control and treatment groups were consistent in their use of antiemetic prophylaxis. The prophylactic use of antiemetics is a routine practice after surgery even in the control group. In the literature we included, 7 explicitly stated the use of prophylactic antiemetics in the postoperative period, 4 stated that antiemetics were not used in the postoperative period, and the remaining 15 were not explicitly described in the literature.

And there were no head to head RCTs of electroacupuncture vs anti-emetics in our included studies. 

5. Figures 2 and 3 are very blurred and hard to read in the main body of the document.

Replay: Thank you for pointing this out. We have re-uploaded the Figures in high resolution.

 

Reviewer #2: The paper entitled Efficacy and safety of electrical acupoint stimulation for postoperative nausea and vomiting: A systematic review and meta-analysis is well designed and well performed meta-analysis. The methods used here are appropriate, the manuscript is clear, well-organized. Only minor remarks are listed below:

- avoid using abbreviations in the abstract (TEAS)

Replay: We appreciate this valuable advice. We deleted all the abbreviations in the abstract.

- lines 74 and 77: repetition

Replay: We apologize for this oversight. This section has been modified to address this point (line 73 to 76).

-line 80: why not sufficiently comprehensive? Explain.

Replay: We are sorry that we didn’t describe this clearly. In our meta-analysis, we included more patients and RCTs. In addition to the inclusion of TEAS, RCTs evaluating the effectiveness of EA in the treatment of PONV were also added. What EA and TEAS have in common is that they both treat diseases by electrical acupoint stimulation. Thus, our study involves a more comprehensive analysis than the 2020 TEAS meta-analysis. We have moved this paragraph from the Introduction to the Discussion and modified the related description in the Discussion and changed the inappropriate statement (line 414 to 418).

- line 84: meanwhile is not the appropriate word.

Replay: We are sorry for the improper use of word. We moved this part to the Discussion and have revised the text (line 418).

- line 172: one category (3 articles) is missing

Replay: Thank you very much for pointing it out. In study selection, we added a category to describe three reports as published in abstract form (line 166).

- line 183: 7 or 8 trials?

Replay: Thank you for your great patience in listing the problems that exist with the manuscript. Here are 8 trials, we have added a Reference in the text (line 185).

- lines 233 AND 237: figure numbering to be corrected

Replay: Thank you for pointed out our mistake. We have modified the numbering of the figures (line 231 to 236).

Thank you again for your careful work.

 

Reviewer #3: Comments to the authors:

1. English language still needs some refinement and editing.

Replay: We thank the reviewer for the comment. The English language has been revised by an expert who is a native speaker.

2. Abbreviation used in the manuscript should have full forms at the time of first mentioning.

Replay: Thank you for pointing this out. We carefully reviewed the revision and made modifications. All of the abbreviations have full forms at the time of first mentioning now.

3. Some details should be discussed about assessment and severity of PONV using Apfel scoring system.

Replay: Thank you very much for your advice. We have added a paragraph in the discussion to describe in detail the assessment and severity of PONV using Apfel scoring system (line 397 to 409).

4. Not only the therapeutic measures, also commonly adopted preventive measures should also me mentioned.

Replay: Thank you so much. Routine use of 5-HT₃ receptor antagonists after surgery is the most common preventive measure. To address this point we added a sentence on the commonly adopted preventive measures in the clinic to the introduction (lines 60 to 65). 

5. Many comments made about EAS are not supported by reference, which should be.

Replay: We agree with the reviewer and the reference has been added to the introduction. Thank you again for your helpful point.

6. Role of EAS outside China is questionable and thus not widely used, as claimed here, even now.

Replay: Thank you for pointing this out. we have removed this description from the introduction. We included 26 studies in 8 countries, so EA is described as increasingly popular in clinical.

7. Uniqueness of your meta-analysis compared to previous ones should be mentioned in the result and discussion sections, not in the introduction section.

Replay: Thanks for the suggestion and we have moved this part from Introduction to the Discussion section (line 410 to 421).

8. Did you use any filter during your literature search?

Replay: No, we did not use any filters when searching the literature. Thank you for your question.

9. What do you mean by “sham” treatment in the control group?

Replay: Compared to the treatment group, sham treatment in the control group included use of an inactive device, no needles with active device, patient-controlled analgesia + inactivated device, patient-controlled analgesia + usual care, which we corrected an inappropriate description in Table I and described the detail in the text (line 182 to 184; line 345 to 347). 

10. Why were cesarean section and abortion excluded?

Replay: Thanks for your question. This is because the abortion patient may have pregnancy vomiting. Similarly, cesarean patients will use oxytocin. The oxytocin cause emesis-producing. And we included patients with general anesthesia, whereas most cesarean patients do not use this type of anesthesia. We describe this in Discussion to make this issue transparent (line 422 to 425). 

11. Regarding the analysis of incidence of PONV, what was used in the control group? Also, you have mentioned incidence with EAS than presenting data separately on EA and TEAS.

Replay: Thank you for asking. The control group included in our study included: use of an inactive device, no needles with active device, no needles with inactive device, patient-controlled analgesia+ inactivated device, patient-controlled analgesia+ usual care. We have now revised this description in the text somewhat to be clearer (line 182 to 184; line 345 to 347). Our aim was to study the role of EAS, therefore, in describing the incidence of PONV, we used EAS as an overall intervention. In the fourth paragraph of the chapter, we did a subgroup analysis to determine whether EA or TEAS had treatment effect. The results showed that both EA and TEAS had significant therapeutic effects.

12. Did you do any subgroup analysis here between different types of surgery or not? Also was the anesthesia technique used standard in all the studies?

Replay: Thank you for your question. First, in our analysis, the surgical types, include laparoscopy, lithotripsy, tonsillectomy, thyroidectomy, gynecological, thoracotomy, craniotomy and plastic surgery. Thus, subgroup analysis according to surgery types will result in a small number of literature grouping.

Second, general anesthesia and standard anesthetic techniques was used in all the studies.

13. What do you mean by “EAS plus other treatments”? What are the other components of the other treatments? From the data I can see the additional other treatment did not decrease the incidence, rather increased the incidence of PONV a bit compared to only EAS. Why so?

Replay: Sorry for the undefined description. "EAS plus other treatments" means that there were other interventions in the treatment group in addition to EAS. Other treatments included the use of PCA in 6 articles and Dexameth in 1 article. Among these 7 studies, these other treatments were also applied in the control group. The variable between two groups in one article was the presence or absence of EAS. We have added an explanation in the "Incidence of PONV" section of our article.

14. In remaining 20% studies where you have mentioned that they received some treatment than placebo in the control arm, what are those interventions used?

Replay: Thank you for your helpful question. As for the control group, some studies used placebo treatment while others did not. Control patients in studies with no placebo treatment were given usual care, while those in studies with placebo treatment recieved either non-electric acupuncture or electrical stimulation to sham acupuncture points besides usual care. We have modified our text in the third part of the Result in order to clarify this point (line 231 to 236). 

15. Apart from the incidence, did you do any analysis of frequency of PONV or not?

Replay: Thank you for your question. We did not do the analysis of frequency of PONV. This is an uncountable component due to the inclusion of very few studies documenting the PONV frequency. In deed, studies including PONV frequencies would make our analysis more credible. 

16. Regarding timing of intervention used, not finding any difference between pre and postoperative EAS means, it does not have any better preemptive effect. Am I right? How do you explain this?

Replay: Thank you for your questions. Unfortunately, we cannot draw that conclusion, using the results obtained in our meta-analysis. Our analysis did not compare the effects of preoperative, intraoperative and postoperative EAS application. Because they are not in the same study. So we were unable to conclude which group had the best efficacy. And there are few such studies at present. In order to get this data, RCTs with higher quality are needed.

17. Targeted and comparative literature review is missing.

Replay: We sincerely appreciate the reviewer's insights. In the Discussion section, we have added a paragraph discussing the development of studies for meta-analysis related to our topic (line 411 to 422). 

Strengths:

• Very relevant topic chosen for meta-analysis.

• Extensive literature search used.

• Extensive statistical analysis and subgroup analysis and multiple outcomes measured.

Weakness:

• English language used is very lucid, has several grammatical mistakes and lacked scientific tone.

• Some design flaws are quite obvious.

• No clarity on control or sham treatment group.

• No clear comparison with already recommended pharmacological treatment group.

• No obvious risk assessment or use of scoring system.

• Several repetitions of statistical methods in the methodology and result section. Especially the result section is confusing.

• The literature review is incomplete without targeted comparative analysis.

Verdict: Good intent of analyzing the efficacy of non-pharmacological measure in very relevant PONV which is still an intriguing issue despite abundance of antiemetics. But the manuscript still needs several modifications, refinement before I can accept it.

---

## [Editor Report · Decision Letter 1]

26 May 2022

PONE-D-21-05426R1

Efficacy and safety of electrical acupoint stimulation for postoperative nausea and vomiting: A systematic review and meta-analysis

PLOS ONE

Dear Dr. Song,

Thank you for submitting your manuscript to PLOS ONE. After careful consideration, we feel that it has merit but does not fully meet PLOS ONE’s publication criteria as it currently stands. Therefore, we invite you to submit a revised version of the manuscript that addresses the points raised during the review process.

We look forward to receiving your revised manuscript.

Kind regards,

Pradipta Bhakta

Guest Editor

PLOS ONE

Additional Editor Comments (if provided):

Comments to the authors:

1. My initial Question: English language still needs some refinement and editing.

Authors’ Reply: We thank the reviewer for the comment. The English language has been revised by an expert who is a native speaker.

My Reply: But I can see there are still many mistakes. Even your reply to my questions has several linguistic and grammatical mistakes. Either you take help from language expert or journal’s service.

2. My initial Question: Abbreviation used in the manuscript should have full forms at the time of first mentioning.

Authors’ Reply: Thank you for pointing this out. We carefully reviewed the revision and made modifications. All of the abbreviations have full forms at the time of first mentioning now.

My Reply: Still many abbreviations are not mentioned in full form initially. Make sure when you reply that you have corrected, you have really address them.

3. My initial Question: Some details should be discussed about assessment and severity of PONV using Apfel scoring system.

Authors’ Reply: Thank you very much for your advice. We have added a paragraph in the discussion to describe in detail the assessment and severity of PONV using Apfel scoring system (line 397 to 409).

My Reply: It should be done in the introduction or methodology, than in the discussion section. In the discussion section, regarding Apfel scoring, you only mentioned general part. Neither have discussed the pros and cons of that, nor have stressed why your own scoring system is superior to that one.

4. My initial Question: Not only the therapeutic measures, also commonly adopted preventive measures should also be mentioned.

Authors’ Reply: Thank you so much. Routine use of 5-HT₃ receptor antagonists after surgery is the most common preventive measure. To address this point we added a sentence on the commonly adopted preventive measures in the clinic to the introduction (lines 60 to 65).

My Reply: Do you think pharmacological mean is the only preventive measure, and nothing else need to be discussed here? Even your comment about routine use of 5-HT3 antagonist is not supported by evidence. You should know that drugs are not only the preventive and therapeutic means for managing PONV.

5. My initial Question: Many comments made about EAS are not supported by reference, which should be.

Authors’ Reply: We agree with the reviewer and the reference has been added to the introduction. Thank you again for your helpful point.

My Reply: Still I can see many comments are not supported by reference. Again, make sure what you reply here, actually you have done that in modified manuscript. No expert comment should be made in the manuscript without citing reference.

6. My initial Question: Role of EAS outside China is questionable and thus not widely used, as claimed here, even now.

Authors’ Reply: Thank you for pointing this out. we have removed this description from the introduction. We included 26 studies in 8 countries, so EA is described as increasingly popular in clinical.

My Reply: Use of EA in 8 countries does not mean universal acceptance. Moreover, can I know which are these 8 countries?

7. My initial Question: Uniqueness of your meta-analysis compared to previous ones should be mentioned in the result and discussion sections, not in the introduction section.

Authors’ Reply: Thanks for the suggestion and we have moved this part from Introduction to the Discussion section (line 410 to 421).

My Reply: Accepted.

8. My initial Question: Did you use any filter during your literature search?

Authors’ Reply: No, we did not use any filters when searching the literature. Thank you for your question.

My Reply: Then how you analysed the searched studies? Did you use only inclusion criteria then?

9. My initial Question: What do you mean by “sham” treatment in the control group?

Authors’ Reply: Compared to the treatment group, sham treatment in the control group included use of an inactive device, no needles with active device, patient-controlled analgesia + inactivated device, patient-controlled analgesia + usual care, which we corrected an inappropriate description in Table I and described the detail in the text (line 182 to 184; line 345 to 347).

My Reply: What do you mean by “inactive device”? Is it dry needling? That is also a treatment, isn’t it?

10. My initial Question: Why were cesarean section and abortion excluded?

Authors’ Reply: Thanks for your question. This is because the abortion patient may have pregnancy vomiting. Similarly, cesarean patients will use oxytocin. The oxytocin cause emesis-producing. And we included patients with general anesthesia, whereas most cesarean patients do not use this type of anesthesia. We describe this in Discussion to make this issue transparent (line 422 to 425).

My Reply: Accepted.

11. My initial Question: Regarding the analysis of incidence of PONV, what was used in the control group? Also, you have mentioned incidence with EAS than presenting data separately on EA and TEAS.

Authors’ Reply: Thank you for asking. The control group included in our study included: use of an inactive device, no needles with active device, no needles with inactive device, patient-controlled analgesia+ inactivated device, patient-controlled analgesia+ usual care. We have now revised this description in the text somewhat to be clearer (line 182 to 184; line 345 to 347). Our aim was to study the role of EAS, therefore, in describing the incidence of PONV, we used EAS as an overall intervention. In the fourth paragraph of the chapter, we did a subgroup analysis to determine whether EA or TEAS had treatment effect. The results showed that both EA and TEAS had significant therapeutic effects.

My Reply: I am still very confused about inactive device with needle. Can you clarify this a bit more?

12. My initial Question: Did you do any subgroup analysis here between different types of surgery or not? Also was the anesthesia technique used standard in all the studies?

Authors’ Reply: Thank you for your question. First, in our analysis, the surgical types, include laparoscopy, lithotripsy, tonsillectomy, thyroidectomy, gynecological, thoracotomy, craniotomy and plastic surgery. Thus, subgroup analysis according to surgery types will result in a small number of literature grouping.

Second, general anesthesia and standard anesthetic techniques was used in all the studies.

My Reply: Lithotripsy, thyroidectomy, thoracotomy, craniotomy, and plastic surgeries are not unique to have high PONV. I agree, that tonsillectomy, gynecological surgeries, laparoscopy are. But why squint surgery was omitted which is a prototype surgery to study PONV?

13. My initial Question: What do you mean by “EAS plus other treatments”? What are the other components of the other treatments? From the data I can see the additional other treatment did not decrease the incidence, rather increased the incidence of PONV a bit compared to only EAS. Why so?

Authors’ Reply: Sorry for the undefined description. "EAS plus other treatments" means that there were other interventions in the treatment group in addition to EAS. Other treatments included the use of PCA in 6 articles and Dexameth in 1 article. Among these 7 studies, these other treatments were also applied in the control group. The variable between two groups in one article was the presence or absence of EAS. We have added an explanation in the "Incidence of PONV" section of our article.

My Reply: Is PCA a treatment option for managing PONV? Dexamethasone has only preventive role in PONV, it is not a therapeutic option. I am really now confused about your open unfiltered search method.

14. My initial Question: In remaining 20% studies where you have mentioned that they received some treatment than placebo in the control arm, what are those interventions used?

Authors’ Reply: Thank you for your helpful question. As for the control group, some studies used placebo treatment while others did not. Control patients in studies with no placebo treatment were given usual care, while those in studies with placebo treatment recieved either non-electric acupuncture or electrical stimulation to sham acupuncture points besides usual care. We have modified our text in the third part of the Result in order to clarify this point (line 231 to 236).

My Reply: Can placebo be called as a treatment option? I am more confused now.

15. My initial Question: Apart from the incidence, did you do any analysis of frequency of PONV or not?

Authors’ Reply: Thank you for your question. We did not do the analysis of frequency of PONV. This is an uncountable component due to the inclusion of very few studies documenting the PONV frequency. Indeed, studies including PONV frequencies would make our analysis more credible.

My Reply: Even if you have not done initially, you need to do it now as this is very important, and this you too have agreed.

16. My initial Question: Regarding timing of intervention used, not finding any difference between pre and postoperative EAS means, it does not have any better pre-emptive effect. Am I right? How do you explain this?

Authors’ Reply: Thank you for your questions. Unfortunately, we cannot draw that conclusion, using the results obtained in our meta-analysis. Our analysis did not compare the effects of preoperative, intraoperative and postoperative EAS application. Because they are not in the same study. So we were unable to conclude which group had the best efficacy. And there are few such studies at present. In order to get this data, RCTs with higher quality are needed.

My Reply: When you did mention that you have done some subgroup analysis and that is the strength of your meta-analysis compared to the 2020 one, then you need to do this.

17. My initial Question: Targeted and comparative literature review is missing.

Authors’ Reply: We sincerely appreciate the reviewer's insights. In the Discussion section, we have added a paragraph discussing the development of studies for meta-analysis related to our topic (line 411 to 422).

My Reply: Targeted literature review does not only mean history of meta-analysis related to your topic and where yours one differ with previous one. This mean analysing the previous literature related to PONV, related to your areas of PONV topic and meta-analysis, and finally where they differed, why they differed from yours meta-analysis. Also, why yours one is unique and what new you have found from them. You have to defend your findings, and negate the differing findings of others.

Strengths:

• Very relevant topic chosen for meta-analysis.

• Extensive literature search used.

• Extensive statistical analysis and subgroup analysis and multiple outcomes measured.

My question: Why no comment here?

Weakness:

• English language used is very lucid, has several grammatical mistakes and lacked scientific tone.

• Some design flaws are quite obvious.

• No clarity on control or sham treatment group.

• No clear comparison with already recommended pharmacological treatment group.

• No obvious risk assessment or use of scoring system.

• Several repetitions of statistical methods in the methodology and result section. Especially the result section is confusing.

• The literature review is incomplete without targeted comparative analysis.

My question: Why no comment here?

Verdict: Good intent of analyzing the efficacy of non-pharmacological measure in very relevant PONV which is still an intriguing issue despite abundance of antiemetics. But the manuscript still needs several modifications, refinement before I can accept it.

My question: Why no comment here?

My New Verdict: Although authors have tried to address the issues I have raised, but unfortunately, many issues still remain unaddressed even though authors have mentioned that they have done. Again considering the importance of the topic and the maturation of the manuscript, I want to give the authors another chance to readdress my points carefully. They should make sure that they have corrected the issues when they mention that they have done. Also they need to do some important subgroup analysis, PONV incidence comparison and should rewrite the literature review and analytic discussion section in a more scientific and argumentative way. If they come up with valid answers to my renewed points and reorganise their manuscript accordingly, I can give another re-look to the manuscript.
---

## [Author Response · Author response to Decision Letter 1]

17 Jul 2022

Dear editor:

On behalf of my co-authors, thank you for giving us another chance to revise our article and improve its quality. We have read the comments carefully by you and other reviewers, and revisions have been done by now. Also, we have tried our best to revise our manuscript according to the comments. Attached please find the revised version, which we would like to submit for your kind consideration. Here, we would like to explain the changes briefly as follows:

1. Initial Question: English language still needs some refinement and editing. 

My Reply: We thank the reviewer for the comment. The English language has been revised by an expert who is a native speaker.

Reviewers’ Reply: But I can see there are still many mistakes. Even your reply to my questions has several linguistic and grammatical mistakes. Either you take help from language expert or journal’s service.

My New Response: We apologize for the linguistic problems in the previous revision and my replies. We worked on the manuscript for a long time, and the repeated addition and removal of sentences obviously led to the previous polishing no longer being applicable. We now have asked American Journal Experts to help us with the language editing in this revision (certificate enclosed). We hope that the language in the current version of the manuscript is acceptable.

2. Initial Question: Abbreviation used in the manuscript should have full forms at the time of first mentioning.

My Reply: Thank you for pointing this out. We carefully reviewed the revision and made modifications. All of the abbreviations have full forms at the time of first mentioning now.

Reviewers’ Reply: Still many abbreviations are not mentioned in full form initially. Make sure when you reply that you have corrected, you have really address them. 

My New Response: We apologize for the abbreviation issue. In this revision, all abbreviations were carefully checked to ensure that they are defined at first use. In addition, we have removed abbreviations that appear only once in our study and replaced them with the full wording. In addition, the abbreviations and their meanings are listed in the following table. Thank you for pointing this out.

Abbreviation Full words Location

PONV Postoperative nausea and vomiting line 51

EA Electroacupuncture line 75

TEAS transcutaneous electrical acupoint stimulation line 76

EAS electrical acupoint stimulation line 78

AMSTAR Assessing the methodological quality of systematic reviews line 94

PON postoperative nausea line 114

POV postoperative vomiting line 114

RCT randomized controlled trials line 115

ASA American Society of Anesthesiologists line 117

GRADE the Grades of Recommendation Assessment Development and Evaluation line 149

RR relative risk line 164

CI confidence interval line 164

3. Initial Question: Some details should be discussed about assessment and severity of PONV using Apfel scoring system.

My Reply: Thank you very much for your advice. We have added a paragraph in the discussion to describe in detail the assessment and severity of PONV using Apfel scoring system (line 397 to 409).

Reviewers’ Reply: It should be done in the introduction or methodology, than in the discussion section. In the discussion section, regarding Apfel scoring, you only mentioned general part. Neither have discussed the pros and cons of that, nor have stressed why your own scoring system is superior to that one.

My New Response: Thank you so much for your questions regarding the scoring system. We apologize for not explaining more clearly why we did not use the Apfel scoring as the main observation in the previous version. The Apfel score is used to predict the risk of PONV but it does not show whether PONV occurred[1]. Our primary outcome index is the incidence of PONV, an objective indicator. Among the included studies, only one article used the Apfel scoring system. It specifies the inclusion of patients with an Apfel score ≥ 2 (the maximum score with this system is 4)[2]. This was done to include people with a high risk for PONV, while none of the other 25 studies had that inclusion criterion. Therefore, based on the current evidence base, we could not include Apfel scores in our analysis. However, we agree that the Apfel scoring system is very important. If these RCTs included Apfel scores in their inclusion criteria or stratified randomized groups of patients based on the Apfel scoring system, higher quality results would have been obtained. We have modified the discussion about Apfel scoring accordingly (lines 414 to 425). 

When you mentioned our “scoring system”, were you referring to the GRADE system? GRADE is a well-established system used to assess the quality of evidence derived from RCTs. It classifies the evidence quality as A (high quality), B (medium quality), C (low quality), or D (very low quality)[3]. Its use is an important, highly recommended step in any comprehensive evidence evaluation and could increase the transparency of the decision-making process [4, 5]. If we misunderstood your question about scoring, please correct us. Thank you again for your helpful suggestion.

4. Initial Question: Not only the therapeutic measures, also commonly adopted preventive measures should also be mentioned.

My Reply: Thank you so much. Routine use of 5-HT₃ receptor antagonists after surgery is the most common preventive measure. To address this point we added a sentence on the commonly adopted preventive measures in the clinic to the introduction (lines 60 to 65).

Reviewers’ Reply: Do you think pharmacological mean is the only preventive measure, and nothing else need to be discussed here? Even your comment about routine use of 5-HT3 antagonist is not supported by evidence. You should know that drugs are not only the preventive and therapeutic means for managing PONV.

My New Response: We apologize for the insufficient clarity in our initial submission. As you mentioned, pharmacological treatment is not the only preventive and therapeutic means for managing PONV. We have added text describing nonpharmacological approaches to PONV prevention in the introduction (lines 68 to 72). To provide a better and more comprehensive description of the prevention and treatment of PONV, we have revised a paragraph in the introduction (lines 60 to 73). Thank you again for your helpful suggestion.

5. Initial Question: Many comments made about EAS are not supported by reference, which should be.

My Reply: We agree with the reviewer and the reference has been added to the introduction. Thank you again for your helpful point.

Reviewers’ Reply: Still I can see many comments are not supported by reference. Again, make sure what you reply here, actually you have done that in modified manuscript. No expert comment should be made in the manuscript without citing reference. 

My New Response: Thank you very much for your careful review. We have carefully checked our manuscript and added 8 references (lines 59, 61, 62, 65, 77, 374 and 409). We hope there is no omission in the revised manuscript.

6. Initial Question: Role of EAS outside China is questionable and thus not widely used, as claimed here, even now.

My Reply: Thank you for pointing this out. we have removed this description from the introduction. We included 26 studies in 8 countries, so EA is described as increasingly popular in clinical.

Reviewers’ Reply: Use of EA in 8 countries does not mean universal acceptance. Moreover, can I know which are these 8 countries? 

My New Response: The 26 studies included in our research were conducted in 8 countries: USA, The United Kingdom, China, India, Malaysia, Turkey, Iran, and Denmark. Table 1 lists the countries for each RCT in our manuscript. As you said, the use of EA in 8 countries does not mean universal acceptance. To make this clearer, we listed the names of these 8 countries in the results (lines 186 to 187). RCTs with higher quality and larger sample sizes need to be performed to promote the usage of EA worldwide. Thank you so much for the suggestion.

7. Initial Question: Uniqueness of your meta-analysis compared to previous ones should be mentioned in the result and discussion sections, not in the introduction section.

My Reply: Thanks for the suggestion and we have moved this part from Introduction to the Discussion section (line 410 to 421).

Reviewers’ Reply: Accepted.

My New Response: Thank you for accepting our reply.

8. Initial Question: Did you use any filter during your literature search?

My Reply: No, we did not use any filters when searching the literature. Thank you for your question.

Reviewers’ Reply: Then how you analysed the searched studies? Did you use only inclusion criteria then? 

My New Response: We apologize for not answering your question clearly. We added more details to our search process, including the literature search strategy, inclusion criteria, and exclusion criteria.

In the first step, using the literature search strategy, we retrieved a total of 864 articles from the databases. In the second step, two researchers independently used EndNote reference management software for literature management. By screening the titles, abstracts, or both and removing duplicates, 794 records were excluded. In the third step, after reading the full articles, we excluded 44 RCTs that met the exclusion criteria. Finally, we included 26 studies.

Literature search strategy (Take PubMed as an example)

#1 Postoperative Nausea and Vomiting [Mesh]

#2 (“post operative” OR “postoperati*” OR “perioperati*” OR “peri-operative” OR “surger*” OR “surgical*” OR “postsurg*” OR “intraoperative” OR “anesthe*” OR “anaesthe*” OR “postanesthe*” OR “postanaesthe*” OR “anaesthetic recovery”) 

#3 (“nause*” OR “vomit*” OR “emesis” OR “emeses” OR “emet*” OR “queasiness” OR “queasy”)

#4 #2 AND #3 

#5 #1 OR #4

#6 Electroacupuncture [Mesh]

#7 electric*

#8 (acupuncture OR needle OR acupoint OR point OR stimulat*)

#9 #7 AND #8

#10 #6 OR #9

#11 Transcutaneous Electric Nerve Stimulation [Mesh]

#12 (“Transcutaneous Electrical Acupoint Stimulation” OR “TENS” OR “TEAS” OR “TNS” OR “ENS” OR “TES” OR “Transcutaneous electric* nerve stimulation” OR “transcutaneous nerve stimulation” OR “transcutaneous electric*” OR “transcutaneous electric* stimulation” OR “electric* nerve therap*” OR “electroanalgesi*” OR “electro-analgesi*” OR “Percutaneous Electric*” OR “Percutaneous Neuromodulation therap*” OR “Electroanalgesia*” OR “nerve stimulat*” OR “neuro-modulation” OR “neuromodulation” OR “neuromusc* electric*”)

#13 #11 OR #12 

#14 Electric Stimulation Therapy [Mesh]

#15 (“Electric Stimulation Therapy” OR “Electric* Stimulation” OR “electrotherap*” OR “electrostimul*” OR “electromyostimulation” OR “Interferential Current Electrotherapy” OR “Therapeutic Electric* stimulat*” OR “Electric* stimulat* therap*”)

#16 #14 OR #15

#17 (“random* controlled trial” OR “random*” OR “placebo”)]

#18 #10 OR #13 OR #16 

#19 #5 AND #18 AND #17

The inclusion criteria were: 

1). Patients who underwent surgery, regardless of age, sex, ethnicity or surgery type 

2). The intervention measures in the treatment group was EAS (EA or TEAS). 

3). The control group could be sham acupuncture, placebo acupuncture, no treatment, nonacupoint acupuncture, no electrical stimulation or perioperative routine nursing. 

4). The outcomes were the incidence of PONV, PON or POV. 

5). To reduce the risk of bias and enhance the accuracy of the conclusions, only studies that had a randomized controlled trials (RCT) design were included. 

The exclusion criteria were as follows:

1). Patients undergoing cesarean section or abortion.

2). American Society of Anesthesiologists ≥ III.

3). Patients in the control group who received any electrical stimulation.

4). Unpublished reports or reports published as abstracts only.

9. Initial Question: What do you mean by “sham” treatment in the control group?

My Reply: Compared to the treatment group, sham treatment in the control group included use of an inactive device, no needles with active device, patient-controlled analgesia + inactivated device, patient-controlled analgesia + usual care, which we corrected an inappropriate description in Table I and described the detail in the text (line 182 to 184; line 345 to 347).

Reviewers’ Reply: What do you mean by “inactive device”? Is it dry needling? That is also a treatment, isn’t it? 

My New Response: We apologize for not clearly describing the different kinds of control treatments. The control treatments include the following four types:

When the treatment is EA, the control treatment includes: no needles + active device; no needles + inactive device or usual care,

When the treatment is TEAS, the control treatment includes: gel electrodes + inactive device or usual care,

no needles + active device: The needles were bent to lay flat against the skin, or no needles were applied. Insulated wires from the activated stimulator box with a normal current output were attached to the needles or the inside of the arm covers. 

no needles + inactive device: The needles were bent to lay flat against the skin, or no needles were applied. Insulated wires from the inactivated stimulator box with no current output were attached to the needles or the inside of the arm covers.

gel electrodes + inactive device: Gel electrodes were placed on the acupoints and attached to a stimulator box with no current output.

Therefore, “no needles + inactive device” is not a dry needling. It has no acupoint stimulation effect. It is a type of placebo. We performed subgroup analysis. The results showed that regardless of what was used as the control, placebo or usual care, EAS showed a significant role in reducing the incidence of PONV (lines 255 to 260). Therefore, we think “no needles + inactive device” should not be counted as a treatment, but only as a kind of psychological comfort. 

10. Initial Question: Why were cesarean section and abortion excluded?

My Reply: Thanks for your question. This is because the abortion patient may have pregnancy vomiting. Similarly, cesarean patients will use oxytocin. The oxytocin cause emesis-producing. And we included patients with general anesthesia, whereas most cesarean patients do not use this type of anesthesia. We describe this in Discussion to make this issue transparent (line 422 to 425).

Reviewers’ Reply: Accepted.

My New Response: It’s very kind of you to accept our answer. Thank you so much.

11. Initial Question: Regarding the analysis of incidence of PONV, what was used in the control group? Also, you have mentioned incidence with EAS than presenting data separately on EA and TEAS.

My Reply: Thank you for asking. The control group included in our study included: use of an inactive device, no needles with active device, no needles with inactive device, patient-controlled analgesia+ inactivated device, patient-controlled analgesia+ usual care. We have now revised this description in the text somewhat to be clearer (line 182 to 184; line 345 to 347). Our aim was to study the role of EAS, therefore, in describing the incidence of PONV, we used EAS as an overall intervention. In the fourth paragraph of the chapter, we did a subgroup analysis to determine whether EA or TEAS had treatment effect. The results showed that both EA and TEAS had significant therapeutic effects.

Reviewers’ Reply: I am still very confused about inactive device with needle. Can you clarify this a bit more? 

My New Response: We apologize for the confusion. In the answer to Question 9, we described all of the controls. The "inactive device with needle" you mentioned here should correspond to "no needle + inactive device", which means that the needles were bent to lay flat against the skin or no needles were applied and insulated wires from the activated stimulator box with no current output were attached to the needles or the insides of arm covers[6, 7]. 

We have revised this description in the text to make this more clear (lines 361 to 369).

12. Initial Question: Did you do any subgroup analysis here between different types of surgery or not? Also was the anesthesia technique used standard in all the studies?

My Reply: Thank you for your question. First, in our analysis, the surgical types, include laparoscopy, lithotripsy, tonsillectomy, thyroidectomy, gynecological, thoracotomy, craniotomy and plastic surgery. Thus, subgroup analysis according to surgery types will result in a small number of literature grouping.

Second, general anesthesia and standard anesthetic techniques was used in all the studies.

Reviewers’ Reply: Lithotripsy, thyroidectomy, thoracotomy, craniotomy, and plastic surgeries are not unique to have high PONV. I agree, that tonsillectomy, gynecological surgeries, laparoscopy are. But why squint surgery was omitted which is a prototype surgery to study PONV? 

My New Response: Thank you for your professional question. Indeed, PONV is the most frequent complication in patients undergoing strabismus surgery[8]. We did not intentionally exclude squint surgery. A search of the literature revealed that these RCTs meeting the inclusion and exclusion criteria did not include squint surgery. 

13. Initial Question: What do you mean by “EAS plus other treatments”? What are the other components of the other treatments? From the data I can see the additional other treatment did not decrease the incidence, rather increased the incidence of PONV a bit compared to only EAS. Why so?

My Reply: Sorry for the undefined description. "EAS plus other treatments" means that there were other interventions in the treatment group in addition to EAS. Other treatments included the use of PCA in 6 articles and Dexameth in 1 article. Among these 7 studies, these other treatments were also applied in the control group. The variable between two groups in one article was the presence or absence of EAS. We have added an explanation in the "Incidence of PONV" section of our article.

Reviewers’ Reply: Is PCA a treatment option for managing PONV? Dexamethasone has only preventive role in PONV, it is not a therapeutic option. I am really now confused about your open unfiltered search method. 

My New Response: Thank you for raising this question. We defined PCA and dexamethasone as “other treatments” inappropriately. PCA is not a treatment option for managing PONV. It is a pain relief method given for postoperative analgesia. PCA is routinely used for postoperative analgesia in some countries. In addition, as you stated, dexamethasone is used as routine prophylaxis, not as a treatment. Six studies explicitly emphasized the use of PCA because all six studies were also about postoperative recovery or postoperative analgesia. One study explicitly emphasized the use of dexamethasone because it focused on the efficacy of dexamethasone and TEAS. Among these 7 studies, PCA/dexamethasone was also applied in the control group. Therefore, neither PCA nor dexamethasone is a specific treatment modality for PONV and is part of the standardized treatment regimen for both groups of patients. The attached table shows the 7 studies that emphasized the usage of PCA/dexamethasone. To remove the ambiguity, we have deleted the "EAS plus other treatments" from the table and text and removed the subgroup analysis of this topic from the results. Thank you again for the helpful feedback and comments on the manuscript.

Author (year) Title of the paper.

El-Rakshy et al, 2009 Effect of intraoperative electroacupuncture on postoperative pain, analgesic requirements, nausea and sedation: a randomised controlled trial.

Gu et al, 2019 The effect of pre-treatment with transcutaneous electrical acupoint stimulation on the quality of recovery after ambulatory breast surgery: a prospective, randomised controlled trial.

Chen et al, 2015 ranscutaneous electric acupoint stimulation alleviates remifentanil-induced hyperalgesia in patients undergoing thyroidectomy: a randomized controlled trial.

Ye et al, 2008 Pain management using Han's acupoint nerve stimulator combined with patient-controlled analgesia following neurosurgery: A randomized case control study

Chen et al, 1998 The effect of location of transcutaneous electrical nerve stimulation on postoperative opioid analgesic requirement: acupoint versus nonacupoint stimulation.

Zheng et al, 2008 Effect of transcutaneous electrical acupoint stimulation on nausea and vomiting induced by patient controlled intravenous analgesia with tramadol.

Yang et al, 2015 Dexamethasone alone vs in combination with transcutaneous electrical acupoint stimulation or tropisetron for prevention of postoperative nausea and vomiting in gynaecological patients undergoing laparoscopic surgery.

14. Initial Question: In remaining 20% studies where you have mentioned that they received some treatment than placebo in the control arm, what are those interventions used?

My Reply: Thank you for your helpful question. As for the control group, some studies used placebo treatment while others did not. Control patients in studies with no placebo treatment were given usual care, while those in studies with placebo treatment recieved either non-electric acupuncture or electrical stimulation to sham acupuncture points besides usual care. We have modified our text in the third part of the Result in order to clarify this point (line 231 to 236).

Reviewers’ Reply: Can placebo be called as a treatment option? I am more confused now. 

My New Response: Except for usual care, the other control treatments can be considered a placebo. We performed subgroup analysis. The results showed that regardless of what was used as the control, placebo or usual care, EAS showed a significant role in reducing the incidence of PONV (lines 255 to 260). We think a placebo should not be counted as a treatment but only as a kind of psychological comfort.

The control treatments include the following four types:

When the treatment is EA, the control treatment includes: no needles + active device, no needles + inactive device or usual care.

When the treatment is TEAS, the control treatment includes: gel electrodes + inactive device and usual care. 

15. Initial Question: Apart from the incidence, did you do any analysis of frequency of PONV or not?

My Reply: Thank you for your question. We did not do the analysis of frequency of PONV. This is an uncountable component due to the inclusion of very few studies documenting the PONV frequency. Indeed, studies including PONV frequencies would make our analysis more credible.

Reviewers’ Reply: Even if you have not done initially, you need to do it now as this is very important, and this you too have agreed. 

My New Response: We apologize for not explaining more clearly why we did not perform an analysis of the frequency of PONV in the previous version of our manuscript. Only one of the 26 studies included in our meta-analysis recorded the frequency of PONV (please see the table below). Therefore, there was insufficient data to conduct a meta-analysis of the frequency of PONV. We have included the lack of data about the frequency of PONV as a limitation of this study (Discussion section lines 473 to 475). We hope this explains why we did not analyze the frequency of PONV.

Author (year) Number of participants

（TEAS / Control） Frequency of PONV

Yeoh et al, 2016 40/40 No record

Kabalak et al, 2005 30/30 TEAS group: 2 times;

Control group: 4 times

Zárate et al, 2001 110/56 No record

Ye et al, 2008 20/20 No record

Chen et al, 1998 25/25 No record

Liu et al, 2008 48/48 No record

An et al, 2014 41/40 No record

Rusy et al, 2002 40/40 No record

Sahmeddini et al, 2010 45/45 No record

El-Rakshy et al, 2009 44/58 No record

Christensen et al, 1989 10/10 No record

Zhang et al, 2014 33/32 No record

Tu et al, 2018 72/72 No record

Tu et al, 2019 60/60 No record

Li et al, 2017 20/20 No record

Gu et al, 2019 58/59 No record

Amir et al, 2007 20/20 No record

Chen et al, 2016 46/46 No record

Yang et al, 2015 50/50 No record

Yu et al, 2020 30/30 No record

Liu et al, 2015 44/44 No record

Chen et al, 2015 29/30 No record

Chen Y et al, 2015 41/42 No record

Yao et al, 2015 35/36 No record

Zheng et al, 2008 30/30 No record

wang et al, 2014 30/30 No record

16. Initial Question: Regarding timing of intervention used, not finding any difference between pre and postoperative EAS means, it does not have any better pre-emptive effect. Am I right? How do you explain this?

My Reply: Thank you for your questions. Unfortunately, we cannot draw that conclusion, using the results obtained in our meta-analysis. Our analysis did not compare the effects of preoperative, intraoperative and postoperative EAS application. Because they are not in the same study. So we were unable to conclude which group had the best efficacy. And there are few such studies at present. In order to get this data, RCTs with higher quality are needed.

Reviewers’ Reply: When you did mention that you have done some subgroup analysis and that is the strength of your meta-analysis compared to the 2020 one, then you need to do this. 

My New Response: We apologize for the unclear explanation in the previous version. To reduce the incidence of PONV, the subgroup analysis showed that EAS, whether applied as a preoperative intervention, postoperative intervention or perioperative intervention, had a significant effect. We can only show that the three types of treatment timing are effective because they were not directly compared in any of the studies.

The sentence “Similarly, no significant subgroup differences were observed among the three types of treatment timing (RR, 0.51; 95% CI, 0.43 to 0.60; P = 0.33)” in the original text means that there was no significant difference in heterogeneity among the three subgroups. We apologize for the confusion. To avoid any further ambiguity, we have removed this sentence.

17. Initial Question: Targeted and comparative literature review is missing.

My Reply: We sincerely appreciate the reviewer's insights. In the Discussion section, we have added a paragraph discussing the development of studies for meta-analysis related to our topic (line 411 to 422).

Reviewers’ Reply: Targeted literature review does not only mean history of meta-analysis related to your topic and where yours one differ with previous one. This mean analysing the previous literature related to PONV, related to your areas of PONV topic and meta-analysis, and finally where they differed, why they differed from yours meta-analysis. Also, why yours one is unique and what new you have found from them. You have to defend your findings, and negate the differing findings of others. 

My New Response: Thank you for the specific suggestion. we rewrote the literature review and analytic discussion section. We have added the targeted and comparative literature to the discussion accordingly (lines 438 to 464).

Thank you for giving us your constructive comments. We think that the revised manuscript has become more convincing.

Strengths:

• Very relevant topic chosen for meta-analysis.

• Extensive literature search used.

• Extensive statistical analysis and subgroup analysis and multiple outcomes measured.

Reviewers’ question: Why no comment here?

My New Response: We apologize for not responding to this comment last time. Thank you very much for your positive appraisal of the manuscript.

Weakness:

• English language used is very lucid, has several grammatical mistakes and lacked scientific tone.

• Some design flaws are quite obvious.

• No clarity on control or sham treatment group.

• No clear comparison with already recommended pharmacological treatment group.

• No obvious risk assessment or use of scoring system.

• Several repetitions of statistical methods in the methodology and result section. Especially the result section is confusing.

• The literature review is incomplete without targeted comparative analysis.

Reviewers’ question: Why no comment here?

My New Response: We apologize for not replying to your comments. Thank you for the reviewer’s careful reading of our manuscript. In light of all the above issues, we have carefully revised the document item by item.

Verdict: Good intent of analyzing the efficacy of non-pharmacological measure in very relevant PONV which is still an intriguing issue despite abundance of antiemetics. But the manuscript still needs several modifications, refinement before I can accept it.

Reviewers’ question: Why no comment here?

My New Response: We appreciate the reviewer’s succinct and accurate summary of our manuscript. Thank you for the kind statements. In light of your questions, we have carefully revised the document item by item. Thank you for taking the time to review our manuscript again. We hope that the current version of our manuscript reaches the standard for acceptance.

Reviewers’ Verdict: Although authors have tried to address the issues I have raised, but unfortunately, many issues still remain unaddressed even though authors have mentioned that they have done. Again considering the importance of the topic and the maturation of the manuscript, I want to give the authors another chance to readdress my points carefully. They should make sure that they have corrected the issues when they mention that they have done. Also they need to do some important subgroup analysis, PONV incidence comparison and should rewrite the literature review and analytic discussion section in a more scientific and argumentative way. If they come up with valid answers to my renewed points and reorganise their manuscript accordingly, I can give another re-look to the manuscript. 

My New Response: Thank you for taking the time to read our revision and giving us your constructive comments. We have made the corresponding changes in the revised manuscript, and we think that the revised manuscript has become more rigorous and convincing. In the answer to the fifteenth question, we explained why we did not analyze the frequency of PONV. In addition, we rewrote the literature review and analytic discussion section. Thank you again for giving us the opportunity to have our manuscript reconsidered. We earnestly appreciate the reviewers’ hard work and hope that the corrections will meet with approval. Should you have any questions, please contact us without hesitation.

Reference

1. Apfel CC, Greim CA, Haubitz I, Goepfert C, Usadel J, Sefrin P, et al. A risk score to predict the probability of postoperative vomiting in adults. Acta anaesthesiologica Scandinavica. 1998;42(5):495-501. Epub 1998/05/30. doi: 10.1111/j.1399-6576.1998.tb05157.x. PubMed PMID: 9605363.

2. Yeoh AH, Tang SS, Abdul Manap N, Wan Mat WR, Said S, Che Hassan MR, et al. Effectiveness of P6 acupoint electrical stimulation in preventing postoperativenausea and vomiting following laparoscopic surgery. Turkish journal of medical sciences. 2016;46(3):620-5. Epub 2016/08/12. doi: 10.3906/sag-1502-56. PubMed PMID: 27513234.

3. Balshem H, Helfand M, Schünemann HJ, Oxman AD, Kunz R, Brozek J, et al. GRADE guidelines: 3. Rating the quality of evidence. Journal of clinical epidemiology. 2011;64(4):401-6. Epub 2011/01/07. doi: 10.1016/j.jclinepi.2010.07.015. PubMed PMID: 21208779.

4. Alonso-Coello P, Schünemann HJ, Moberg J, Brignardello-Petersen R, Akl EA, Davoli M, et al. GRADE Evidence to Decision (EtD) frameworks: a systematic and transparent approach to making well informed healthcare choices. 1: Introduction. BMJ (Clinical research ed). 2016;353:i2016. Epub 2016/06/30. doi: 10.1136/bmj.i2016. PubMed PMID: 27353417.

5. Xin Z, Xue-Ting L, De-Ying K. GRADE in Systematic Reviews of Acupuncture for Stroke Rehabilitation: Recommendations based on High-Quality Evidence. Scientific reports. 2015;5:16582. Epub 2015/11/13. doi: 10.1038/srep16582. PubMed PMID: 26560971; PubMed Central PMCID: PMCPMC4642304.

6. Rusy LM, Hoffman GM, Weisman SJ. Electroacupuncture prophylaxis of postoperative nausea and vomiting following pediatric tonsillectomy with or without adenoidectomy. Anesthesiology. 2002;96(2):300-5. Epub 2002/01/31. doi: 10.1097/00000542-200202000-00013. PubMed PMID: 11818760.

7. Sahmeddini MA, Farbood A, Ghafaripuor S. Electro-acupuncture for pain relief after nasal septoplasty: a randomized controlled study. Journal of alternative and complementary medicine (New York, NY). 2010;16(1):53-7. Epub 2009/12/17. doi: 10.1089/acm.2009.0288. PubMed PMID: 20001536.

8. Sayed JA, MA FR, MO MA. Comparison of dexamethasone or intravenous fluids or combination of both on postoperative nausea, vomiting and pain in pediatric strabismus surgery. Journal of clinical anesthesia. 2016;34:136-42. Epub 2016/10/01. doi: 10.1016/j.jclinane.2016.03.049. PubMed PMID: 27687360.

---

## [Decision Letter · Decision Letter 2]

24 Nov 2022

PONE-D-21-05426R2Efficacy and safety of electrical acupoint stimulation for postoperative nausea and vomiting: A systematic review and meta-analysisPLOS ONE

Dear Dr. Song,

Thank you for submitting your manuscript to PLOS ONE. After careful consideration, we feel that it has merit but does not fully meet PLOS ONE’s publication criteria as it currently stands. Therefore, we invite you to submit a revised version of the manuscript that addresses the points raised during the review process.

We look forward to receiving your revised manuscript.

Kind regards,

Huijuan Cao, Ph.D.

Academic Editor

PLOS ONE

Additional Editor Comments:

The English of this article needs to be further improved. Professional language polishing team can be considered for assistance

Reviewers' comments:

Reviewer's Responses to Questions

**Comments to the Author**

1. If the authors have adequately addressed your comments raised in a previous round of review and you feel that this manuscript is now acceptable for publication, you may indicate that here to bypass the “Comments to the Author” section, enter your conflict of interest statement in the “Confidential to Editor” section, and submit your "Accept" recommendation.

Reviewer #3: All comments have been addressed

Reviewer #4: All comments have been addressed

2. Is the manuscript technically sound, and do the data support the conclusions?

Reviewer #3: Partly

Reviewer #4: Partly

3. Has the statistical analysis been performed appropriately and rigorously? 

Reviewer #3: Yes

Reviewer #4: Yes

4. Have the authors made all data underlying the findings in their manuscript fully available?

Reviewer #3: Yes

Reviewer #4: Yes

5. Is the manuscript presented in an intelligible fashion and written in standard English?

Reviewer #3: No

Reviewer #4: Yes

6. Review Comments to the Author

Reviewer #3: My Comments to the authors’ revised version:

1. My initial Question: English language still needs some refinement and editing.

Authors’ Repeat Response: We apologize for the linguistic problems in the previous revision and my replies. We worked on the manuscript for a long time, and the repeated addition and removal of sentences obviously led to the previous polishing no longer being applicable. We now have asked American Journal Experts to help us with the language editing in this revision (certificate enclosed). We hope that the language in the current version of the manuscript is acceptable.

My repeat response: I can still see several grammatical mistake, mistakes in sentence organisation as well as use of punctuation. I cannot do English editing repeatedly, and neither have time to do that. I leave this to the handling editor. If done properly, then OK for me.

2. My initial Question: Abbreviation used in the manuscript should have full forms at the time of first mentioning.

Authors’ Repeat Response: We apologize for the abbreviation issue. In this revision, all abbreviations were carefully checked to ensure that they are defined at first use. In addition, we have removed abbreviations that appear only once in our study and replaced them with the full wording. In addition, the abbreviations and their meanings are listed in the following table. Thank you for pointing this out.

My repeat response: Accepted if all corrected.

3. My initial Question: Some details should be discussed about assessment and severity of PONV using Apfel scoring system.

Authors’ Repeat Response: Thank you so much for your questions regarding the scoring system. We apologize for not explaining more clearly why we did not use the Apfel scoring as the main observation in the previous version. The Apfel score is used to predict the risk of PONV but it does not show whether PONV occurred[1]. Our primary outcome index is the incidence of PONV, an objective indicator. Among the included studies, only one article used the Apfel scoring system. It specifies the inclusion of patients with an Apfel score ≥ 2 (the maximum score with this system is 4)[2]. This was done to include people with a high risk for PONV, while none of the other 25 studies had that inclusion criterion. Therefore, based on the current evidence base, we could not include Apfel scores in our analysis. However, we agree that the Apfel scoring system is very important. If these RCTs included Apfel scores in their inclusion criteria or stratified randomized groups of patients based on the Apfel scoring system, higher quality results would have been obtained. We have modified the discussion about Apfel scoring accordingly (lines 414 to 425).

When you mentioned our “scoring system”, were you referring to the GRADE system? GRADE is a well-established system used to assess the quality of evidence derived from RCTs. It classifies the evidence quality as A (high quality), B (medium quality), C (low quality), or D (very low quality)[3]. Its use is an important, highly recommended step in any comprehensive evidence evaluation and could increase the transparency of the decision-making process [4, 5]. If we misunderstood your question about scoring, please correct us. Thank you again for your helpful suggestion.

My repeat response: Agree that Apfel scoring system identifies the at risk patients preoperatively, but it does predict the PONV as well. Although you have again mentioned the risk of PONV, but there is no mention how to evaluate and stratify those risks. This is confusing.

4. My initial Question: Not only the therapeutic measures, also commonly adopted preventive measures should also be mentioned.

Authors’ Repeat Response: We apologize for the insufficient clarity in our initial submission. As you mentioned, pharmacological treatment is not the only preventive and therapeutic means for managing PONV. We have added text describing nonpharmacological approaches to PONV prevention in the introduction (lines 68 to 72). To provide a better and more comprehensive description of the prevention and treatment of PONV, we have revised a paragraph in the introduction (lines 60 to 73). Thank you again for your helpful suggestion.

My repeat response: Good for this acknowledgement. What I wanted to mean is that along with non-pharmacological means and modification of patient’s factors as well as the anaesthesia technique, using dexamethasone in the beginning and 5-HT3 antagonist at the end based on Apfel Risk score is the best method to prevent PONV. This should be clearly mentioned. Again in the non-pharmacological means you have mentioned crystalloids, chewing gum, ginger etc. These are indeed pharmacological means. Pharmacological means doesn’t have to me classic antiemetic medicine, but can be traditional medicines as well including IV fluids.

5. My initial Question: Many comments made about EAS are not supported by reference, which should be.

Authors’ Repeat Response: Thank you very much for your careful review. We have carefully checked our manuscript and added 8 references (lines 59, 61, 62, 65, 77, 374 and 409). We hope there is no omission in the revised manuscript.

My repeat response: Accepted.

6. My initial Question: Role of EAS outside China is questionable and thus not widely used, as claimed here, even now.

Authors’ Repeat Response: The 26 studies included in our research were conducted in 8 countries: USA, The United Kingdom, China, India, Malaysia, Turkey, Iran, and Denmark. Table 1 lists the countries for each RCT in our manuscript. As you said, the use of EA in 8 countries does not mean universal acceptance. To make this clearer, we listed the names of these 8 countries in the results (lines 186 to 187). RCTs with higher quality and larger sample sizes need to be performed to promote the usage of EA worldwide. Thank you so much for the suggestion.

My repeat response: Accepted.

7. My initial Question: Uniqueness of your meta-analysis compared to previous ones should be mentioned in the result and discussion sections, not in the introduction section.

Authors’ Reply: Thanks for the suggestion and we have moved this part from Introduction to the Discussion section (line 410 to 421).

My Response: Accepted.

8. My initial Question: Did you use any filter during your literature search?

Authors’ Repeat Response: We apologize for not answering your question clearly. We added more details to our search process, including the literature search strategy, inclusion criteria, and exclusion criteria. In the first step, using the literature search strategy, we retrieved a total of 864 articles from the databases. In the second step, two researchers independently used EndNote reference management software for literature management. By screening the titles, abstracts, or both and removing duplicates, 794 records were excluded. In the third step, after reading the full articles, we excluded 44 RCTs that met the exclusion criteria. Finally, we included 26 studies.

My repeat response: Accepted. The sentence of the inclusion and exclusion criteria should be revamped and organised to make it look more scientific.

9. My initial Question: What do you mean by “sham” treatment in the control group?

Authors’ Repeat Response: We apologize for not clearly describing the different kinds of control treatments. The control treatments include the following four types: When the treatment is EA, the control treatment includes: no needles + active device; no needles + inactive device or usual care, When the treatment is TEAS, the control treatment includes: gel electrodes + inactive device or usual care, no needles + active device: The needles were bent to lay flat against the skin, or no needles were applied. Insulated wires from the activated stimulator box with a normal current output were attached to the needles or the inside of the arm covers. no needles + inactive device: The needles were bent to lay flat against the skin, or no needles were applied. Insulated wires from the inactivated stimulator box with no current output were attached to the needles or the inside of the arm covers. gel electrodes + inactive device: Gel electrodes were placed on the acupoints and attached to a stimulator box with no current output. Therefore, “no needles + inactive device” is not a dry needling. It has no acupoint stimulation effect. It is a type of placebo. We performed subgroup analysis. The results showed that regardless of what was used as the control, placebo or usual care, EAS showed a significant role in reducing the incidence of PONV (lines 255 to 260). Therefore, we think “no needles + inactive device” should not be counted as a treatment, but only as a kind of psychological comfort.

My repeat response: I am a bit confused here. What did you exactly mean by “active” and “inactive” device with or without needle? In these complex and mixed methodology how did you do the double blinding?

10. My initial Question: Why were cesarean section and abortion excluded?

Authors’ Repeat Response: It’s very kind of you to accept our answer. Thank you so much.

11. My initial Question: Regarding the analysis of incidence of PONV, what was used in the control group? Also, you have mentioned incidence with EAS than presenting data separately on EA and TEAS.

Authors’ Repeat Response: We apologize for the confusion. In the answer to Question 9, we described all of the controls. The "inactive device with needle" you mentioned here should correspond to "no needle + inactive device", which means that the needles were bent to lay flat against the skin or no needles were applied and insulated wires from the activated stimulator box with no current output were attached to the needles or the insides of arm covers [6, 7]. We have revised this description in the text to make this more clear (lines 361 to 369).

My repeat response: What is the active device without needle then?

12. My initial Question: Did you do any subgroup analysis here between different types of surgery or not? Also was the anesthesia technique used standard in all the studies?

Authors’ Repeat Response: Thank you for your professional question. Indeed, PONV is the most frequent complication in patients undergoing strabismus surgery[8]. We did not intentionally exclude squint surgery. A search of the literature revealed that these RCTs meeting the inclusion and exclusion criteria did not include squint surgery.

My repeat response: Surprising, but accepted.

13. My initial Question: What do you mean by “EAS plus other treatments”? What are the other components of the other treatments? From the data I can see the additional other treatment did not decrease the incidence, rather increased the incidence of PONV a bit compared to only EAS. Why so?

Authors’ Repeat Response: Thank you for raising this question. We defined PCA and dexamethasone as “other treatments” inappropriately. PCA is not a treatment option for managing PONV. It is a pain relief method given for postoperative analgesia. PCA is routinely used for postoperative analgesia in some countries. In addition, as you stated, dexamethasone is used as routine prophylaxis, not as a treatment. Six studies explicitly emphasized the use of PCA because all six studies were also about postoperative recovery or postoperative analgesia. One study explicitly emphasized the use of dexamethasone because it focused on the efficacy of dexamethasone and TEAS. Among these 7 studies, PCA/dexamethasone was also applied in the control group. Therefore, neither PCA nor dexamethasone is a specific treatment modality for PONV and is part of the standardized treatment regimen for both groups of patients. The attached table shows the 7 studies that emphasized the usage of PCA/dexamethasone. To remove the ambiguity, we have deleted the "EAS plus other treatments" from the table and text and removed the subgroup analysis of this topic from the results. Thank you again for the helpful feedback and comments on the manuscript.

My repeat response: Partially agree. Even though your included studies have used PCA for postoperative analgesia (possibly opioid), it should not be included as measure for PONV, rather will be a proponent of causing PONV. Yes, I agree that prophylactic dexamethasone can really have in impact on the incidence of PONV and thus should be included in the meta-analysis.

14. My initial Question: In remaining 20% studies where you have mentioned that they received some treatment than placebo in the control arm, what are those interventions used?

Authors’ Repeat Response: Except for usual care, the other control treatments can be considered a placebo. We performed subgroup analysis. The results showed that regardless of what was used as the control, placebo or usual care, EAS showed a significant role in reducing the incidence of PONV (lines 255 to 260). We think a placebo should not be counted as a treatment but only as a kind of psychological comfort.

The control treatments include the following four types:

When the treatment is EA: the control treatment included no needles + active device, no needles + inactive device or usual care.

When the treatment is TEAS: the control treatment included gel electrodes + inactive device and usual care.

My repeat response: Although I agree ultimately, but I am not convinced with your explanation that placebo can be a treatment option. Usually placebo means no treatment. Control group can have usual care when you are comparing with a new care like EAS and TEAS.

15. My initial Question: Apart from the incidence, did you do any analysis of frequency of PONV or not?

Authors’ Repeat Response: We apologize for not explaining more clearly why we did not perform an analysis of the frequency of PONV in the previous version of our manuscript. Only one of the 26 studies included in our meta-analysis recorded the frequency of PONV (please see the table below). Therefore, there was insufficient data to conduct a meta-analysis of the frequency of PONV. We have included the lack of data about the frequency of PONV as a limitation of this study (Discussion section lines 473 to 475). We hope this explains why we did not analyze the frequency of PONV.

My repeat response: Although not fully happy, but agree.

16. My initial Question: Regarding timing of intervention used, not finding any difference between pre and postoperative EAS means, it does not have any better pre-emptive effect. Am I right? How do you explain this?

Authors’ Repeat Response: We apologize for the unclear explanation in the previous version. To reduce the incidence of PONV, the subgroup analysis showed that EAS, whether applied as a preoperative intervention, postoperative intervention or perioperative intervention, had a significant effect. We can only show that the three types of treatment timing are effective because they were not directly compared in any of the studies.

The sentence “Similarly, no significant subgroup differences were observed among the three types of treatment timing (RR, 0.51; 95% CI, 0.43 to 0.60; P = 0.33)” in the original text means that there was no significant difference in heterogeneity among the three subgroups. We apologize for the confusion. To avoid any further ambiguity, we have removed this sentence.

My repeat response: Accepted.

17. My initial Question: Targeted and comparative literature review is missing.

Authors’ Repeat Response: Thank you for the specific suggestion. we rewrote the literature review and analytic discussion section. We have added the targeted and comparative literature to the discussion accordingly (lines 438 to 464). Thank you for giving us your constructive comments. We think that the revised manuscript has become more convincing.

My repeat response: Again I am unable to see targeted comparative literature review. For your every outcome measures you should mention some previous literature whether it is matching or you have found a new one. If not matching, why so. Simple descriptive literature review is unnecessary here.

General discussion on PONV and its etiology are immaterial here as this is very well-known. Keep the literature review pertaining to your topic of interest.

Weakness:

• English language used is very lucid, has several grammatical mistakes and lacked scientific tone.

• Some design flaws are quite obvious.

• No clarity on control or sham treatment group.

• No clear comparison with already recommended pharmacological treatment group.

• No obvious risk assessment or use of scoring system.

• Several repetitions of statistical methods in the methodology and result section. Especially the result section is confusing.

• The literature review is incomplete without targeted comparative analysis.

Authors’ Reply: We apologize for not replying to your comments. Thank you for the reviewer’s careful reading of our manuscript. In light of all the above issues, we have carefully revised the document item by item.

My repeat response: Your all loopholes and gaps need to be highlighted in the limitation section clearly.

Discussion should be started with findings of your meta-analysis than the group description. The description of groups with active or inactive with or without needles should come in the methodology or literature search section, not in the first paragraph of discussion.

My Initial impression: Good intent of analyzing the efficacy of non-pharmacological measures in very relevant PONV topic which is still an intriguing issue despite abundance of availability of antiemetics. But the manuscript still needs several modifications, refinement before I can accept it.

Authors’ Reply: Thank you for taking the time to read our revision and giving us your constructive comments. We have made the corresponding changes in the revised manuscript, and we think that the revised manuscript has become more rigorous and convincing. In the answer to the fifteenth question, we explained why we did not analyze the frequency of PONV. In addition, we rewrote the literature review and analytic discussion section. Thank you again for giving us the opportunity to have our manuscript reconsidered. We earnestly appreciate the reviewers’ hard work and hope that the corrections will meet with approval. Should you have any questions, please contact us without hesitation.

My Renewed Impression: Although the manuscript has matured much, but still there are areas which need to be improved further including the English language as well as targeted comparative literature review. I shall like to give authors another chance to improve the manuscript as it is improving in right direction. If they come up with clear explanations on the points I have raised and modify the manuscript accordingly I will give another look.

Reviewer #4: 1. Please describe the treatment methods, including the treatment site, treatment schedule, electrical stimulation intensity, evaluation schedule, and the results of RCTs included in this study in more detail.

2. Please provide the literature search strategy of each electronic database as a additional file in a supporting information.

3. Although the quality of evidence was moderate to low, the conclusion in the abstract section is too positive. Please revise it.

4. The authors provided reference 32 and 33 as the basis for the sentence “ EAS is becoming increasingly popular in clinical practice due to its repeatability and standardization of frequency, intensity and duration “ in the introduction section ( page 4, line 82-83). I wonder how this sentence relates to reference 32 and 33.

7. PLOS authors have the option to publish the peer review history of their article (what does this mean?). If published, this will include your full peer review and any attached files.

Reviewer #3: **Yes: **Pradipta Bhakta

Reviewer #4: No

---

## [Author Response · Author response to Decision Letter 2]

4 Jan 2023

Dear Editors and Reviewers: 

Thank you very much again for giving us another chance to revise our article. And We are very grateful for your valuable suggestions and comments of the paper that are very helpful for us to improve our work. We have tried our best to revise our manuscript according to the comments. We are very happy that the reviewers accepted our explanation to some questions. We are now removing the questions that were accepted last time, leaving the questions that still need to be answered. We hope that your suggestions and comments could be perfectly addressed and the revised manuscript could meet the requirements for publication.

Attached please find the revised version, which we would like to submit for your kind consideration. We below present our responses and indicate where new information can be found in the original manuscript.

1. Initial Question: English language still needs some refinement and editing. 

My Reply: We thank the reviewer for the comment. The English language has been revised by an expert who is a native speaker.

Reviewers’ Reply: But I can see there are still many mistakes. Even your reply to my questions has several linguistic and grammatical mistakes. Either you take help from language expert or journal’s service.

My last Response: We apologize for the linguistic problems in the previous revision and my replies. We worked on the manuscript for a long time, and the repeated addition and removal of sentences obviously led to the previous polishing no longer being applicable. We now have asked American Journal Experts to help us with the language editing in this revision (certificate enclosed). We hope that the language in the current version of the manuscript is acceptable.

Reviewers’ repeat response: I can still see several grammatical mistake, mistakes in sentence organisation as well as use of punctuation. I cannot do English editing repeatedly, and neither have time to do that. I leave this to the handling editor. If done properly, then OK for me.

My New Response: We thank you for the patience and suggestion. We have seriously revised our manuscript again. This has been done to the best of our abilities. We will appreciate if editor could help us to improve our manuscript writing.

3. Initial Question: Some details should be discussed about assessment and severity of PONV using Apfel scoring system.

My Reply: Thank you very much for your advice. We have added a paragraph in the discussion to describe in detail the assessment and severity of PONV using Apfel scoring system (line 397 to 409).

Reviewers’ Reply: It should be done in the introduction or methodology, than in the discussion section. In the discussion section, regarding Apfel scoring, you only mentioned general part. Neither have discussed the pros and cons of that, nor have stressed why your own scoring system is superior to that one.

My last Response: Thank you so much for your questions regarding the scoring system. We apologize for not explaining more clearly why we did not use the Apfel scoring as the main observation in the previous version. The Apfel score is used to predict the risk of PONV but it does not show whether PONV occurred[1]. Our primary outcome index is the incidence of PONV, an objective indicator. Among the included studies, only one article used the Apfel scoring system. It specifies the inclusion of patients with an Apfel score ≥ 2 (the maximum score with this system is 4)[2]. This was done to include people with a high risk for PONV, while none of the other 25 studies had that inclusion criterion. Therefore, based on the current evidence base, we could not include Apfel scores in our analysis. However, we agree that the Apfel scoring system is very important. If these RCTs included Apfel scores in their inclusion criteria or stratified randomized groups of patients based on the Apfel scoring system, higher quality results would have been obtained. We have modified the discussion about Apfel scoring accordingly (lines 414 to 425). 

When you mentioned our “scoring system”, were you referring to the GRADE system? GRADE is a well-established system used to assess the quality of evidence derived from RCTs. It classifies the evidence quality as A (high quality), B (medium quality), C (low quality), or D (very low quality)[3]. Its use is an important, highly recommended step in any comprehensive evidence evaluation and could increase the transparency of the decision-making process [4, 5]. If we misunderstood your question about scoring, please correct us. Thank you again for your helpful suggestion.

Reviewers’ repeat response: Agree that Apfel scoring system identifies the at risk patients preoperatively, but it does predict the PONV as well. Although you have again mentioned the risk of PONV, but there is no mention how to evaluate and stratify those risks. This is confusing.

My New Response: Thank you again for your questions regarding the scoring system. In order to evaluate whether EAS is useful for patients with “low,” “medium,” or “high” risk of PONV, RCTs in which participants were stratified by Apfel score should be done. However, only one article mentioned the Apfel scoring system during patient inclusion among the included studies. It specifies the inclusion of patients with an Apfel score ≥ 2 (the maximum score with this system is 4) including patients with a medium-high risk for PONV [2]. This was done to include people with a medium or high risk for PONV, while none of the other 25 studies had that inclusion criterion. Thus, we could not conduct a subgroup analysis based on Apfel scores. We have added the discussion of how to evaluate and stratify the risk of PONV (Discussion section lines 454 to 468). And we have included the lack of scoring system as a limitation of this study (Discussion section lines 488 to 489).

4. Initial Question: Not only the therapeutic measures, also commonly adopted preventive measures should also be mentioned.

My Reply: Thank you so much. Routine use of 5-HT₃ receptor antagonists after surgery is the most common preventive measure. To address this point we added a sentence on the commonly adopted preventive measures in the clinic to the introduction (lines 60 to 65).

Reviewers’ Reply: Do you think pharmacological mean is the only preventive measure, and nothing else need to be discussed here? Even your comment about routine use of 5-HT3 antagonist is not supported by evidence. You should know that drugs are not only the preventive and therapeutic means for managing PONV.

My last Response: We apologize for the insufficient clarity in our initial submission. As you mentioned, pharmacological treatment is not the only preventive and therapeutic means for managing PONV. We have added text describing nonpharmacological approaches to PONV prevention in the introduction (lines 68 to 72). To provide a better and more comprehensive description of the prevention and treatment of PONV, we have revised a paragraph in the introduction (lines 60 to 73). Thank you again for your helpful suggestion.

Reviewers’ repeat response: Good for this acknowledgement. What I wanted to mean is that along with non-pharmacological means and modification of patient’s factors as well as the anaesthesia technique, using dexamethasone in the beginning and 5-HT3 antagonist at the end based on Apfel Risk score is the best method to prevent PONV. This should be clearly mentioned. Again in the non-pharmacological means you have mentioned crystalloids, chewing gum, ginger etc. These are indeed pharmacological means. Pharmacological means doesn’t have to me classic antiemetic medicine, but can be traditional medicines as well including IV fluids.

My New Response: Thank you very much for your precious reminding. According to your indication, we add the following sentence to the introduction section: Using dexamethasone in the beginning and 5-HT3 antagonist at the end based on Apfel Risk score is the best method to prevent PONV (Lines 64 to 66). As you pointed out, crystalloids, chewing gum, ginger are indeed pharmacological means. We made the correction (Lines 65 to 67; Lines 70 to 72).

8. Initial Question: Did you use any filter during your literature search?

My Reply: No, we did not use any filters when searching the literature. Thank you for your question.

Reviewers’ Reply: Then how you analysed the searched studies? Did you use only inclusion criteria then? 

My last Response: We apologize for not answering your question clearly. We added more details to our search process, including the literature search strategy, inclusion criteria, and exclusion criteria.

In the first step, using the literature search strategy, we retrieved a total of 864 articles from the databases. In the second step, two researchers independently used EndNote reference management software for literature management. By screening the titles, abstracts, or both and removing duplicates, 794 records were excluded. In the third step, after reading the full articles, we excluded 44 RCTs that met the exclusion criteria. Finally, we included 26 studies.

Literature search strategy (Take PubMed as an example)

#1 Postoperative Nausea and Vomiting [Mesh]

#2 (“post operative” OR “postoperati*” OR “perioperati*” OR “peri-operative” OR “surger*” OR “surgical*” OR “postsurg*” OR “intraoperative” OR “anesthe*” OR “anaesthe*” OR “postanesthe*” OR “postanaesthe*” OR “anaesthetic recovery”) 

#3 (“nause*” OR “vomit*” OR “emesis” OR “emeses” OR “emet*” OR “queasiness” OR “queasy”)

#4 #2 AND #3 

#5 #1 OR #4

#6 Electroacupuncture [Mesh]

#7 electric*

#8 (acupuncture OR needle OR acupoint OR point OR stimulat*)

#9 #7 AND #8

#10 #6 OR #9

#11 Transcutaneous Electric Nerve Stimulation [Mesh]

#12 (“Transcutaneous Electrical Acupoint Stimulation” OR “TENS” OR “TEAS” OR “TNS” OR “ENS” OR “TES” OR “Transcutaneous electric* nerve stimulation” OR “transcutaneous nerve stimulation” OR “transcutaneous electric*” OR “transcutaneous electric* stimulation” OR “electric* nerve therap*” OR “electroanalgesi*” OR “electro-analgesi*” OR “Percutaneous Electric*” OR “Percutaneous Neuromodulation therap*” OR “Electroanalgesia*” OR “nerve stimulat*” OR “neuro-modulation” OR “neuromodulation” OR “neuromusc* electric*”)

#13 #11 OR #12 

#14 Electric Stimulation Therapy [Mesh]

#15 (“Electric Stimulation Therapy” OR “Electric* Stimulation” OR “electrotherap*” OR “electrostimul*” OR “electromyostimulation” OR “Interferential Current Electrotherapy” OR “Therapeutic Electric* stimulat*” OR “Electric* stimulat* therap*”)

#16 #14 OR #15

#17 (“random* controlled trial” OR “random*” OR “placebo”)]

#18 #10 OR #13 OR #16 

#19 #5 AND #18 AND #17

The inclusion criteria were: 

1). Patients who underwent surgery, regardless of age, sex, ethnicity or surgery type 

2). The intervention measures in the treatment group was EAS (EA or TEAS). 

3). The control group could be sham acupuncture, placebo acupuncture, no treatment, nonacupoint acupuncture, no electrical stimulation or perioperative routine nursing. 

4). The outcomes were the incidence of PONV, PON or POV. 

5). To reduce the risk of bias and enhance the accuracy of the conclusions, only studies that had a randomized controlled trials (RCT) design were included. 

The exclusion criteria were as follows:

1). Patients undergoing cesarean section or abortion.

2). American Society of Anesthesiologists ≥ III.

3). Patients in the control group who received any electrical stimulation.

4). Unpublished reports or reports published as abstracts only.

Reviewers’ repeat response: Accepted. The sentence of the inclusion and exclusion criteria should be revamped and organised to make it look more scientific.

My New Response: Thank you for your kind accepting and your suggestion. We have revamped and organized the inclusion and exclusion criteria (Lines 119 to 130).

The inclusion criteria were: 

1. The review included emergency and elective surgery patients that underwent general anesthesia, regardless of age, sex, ethnicity, or surgery type. 

2. The intervention measure in the treatment group was EAS (EA or TEAS). 

3. the intervention measures in the control group were sham acupuncture, sham acupoint, sham electrical stimulation, or preoperative routine nursing. 

4. Primary outcomes were the incidence of PONV, postoperative nausea (PON) or postoperative vomiting (POV). Secondary outcome measures were the numbers of patients requiring antiemetic rescue and adverse events. 

5. The study was a randomized controlled trial (RCT) (blinded or non-blinded).

The exclusion criteria were as follows: 

1. Patients undergoing cesarean section or abortion or those in the control group who received any electrical stimulation. 

2. American Society of Anesthesiologists (ASA) ≥ III. 

3. Literature with incorrect data or inaccessible data.

9. Initial Question: What do you mean by “sham” treatment in the control group?

My Reply: Compared to the treatment group, sham treatment in the control group included use of an inactive device, no needles with active device, patient-controlled analgesia + inactivated device, patient-controlled analgesia + usual care, which we corrected an inappropriate description in Table I and described the detail in the text (line 182 to 184; line 345 to 347).

Reviewers’ Reply: What do you mean by “inactive device”? Is it dry needling? That is also a treatment, isn’t it? 

My New Response: We apologize for not clearly describing the different kinds of control treatments. The control treatments include the following four types:

When the treatment is EA, the control treatment includes: no needles + active device; no needles + inactive device or usual care,

When the treatment is TEAS, the control treatment includes: gel electrodes + inactive device or usual care,

no needles + active device: The needles were bent to lay flat against the skin, or no needles were applied. Insulated wires from the activated stimulator box with a normal current output were attached to the needles or the inside of the arm covers. 

no needles + inactive device: The needles were bent to lay flat against the skin, or no needles were applied. Insulated wires from the inactivated stimulator box with no current output were attached to the needles or the inside of the arm covers.

gel electrodes + inactive device: Gel electrodes were placed on the acupoints and attached to a stimulator box with no current output.

Therefore, “no needles + inactive device” is not a dry needling. It has no acupoint stimulation effect. It is a type of placebo. We performed subgroup analysis. The results showed that regardless of what was used as the control, placebo or usual care, EAS showed a significant role in reducing the incidence of PONV (lines 255 to 260). Therefore, we think “no needles + inactive device” should not be counted as a treatment, but only as a kind of psychological comfort. 

Reviewers’ repeat response: I am a bit confused here. What did you exactly mean by “active” and “inactive” device with or without needle? In these complex and mixed methodology how did you do the double blinding?

My New Response: For the control treatments part, we are sorry for our confusing descriptions. Now, we have organized our mind so that we can be much easier to explain. In order to avoid any ambiguity, we rename “no needles” into “no acupuncture” (Lines: 106 to 107, 110, 216, and 271 to 272). The details of changes are as follows:

1. “active device” means Insulated wires from the activated stimulator box with a normal current output;

2. “Inactive device” means Insulated wires from the inactivated stimulator box with no current output (for example, the output wires of the stimulator were broken)

3. No needles essentially mean no acupuncture, include the following two types:

with needle: The needles were bent to lay flat against the skin;

without needle: the wire of the stimulator is pasted over the hand without needle and stimulation;

For the blinding part, we evaluated the reasonableness of the blinding method from the description of the 26 included studies. If the blinding was unclear, we marked it in yellow in our evaluation. If the blinding was not appropriate, we mark red in the evaluation (see S1 Fig). We apologize for not clearly describing the blinding. We think part of the articles are possible to be double blind, for the patients and outcome assessor. Considering the specificity of EAS treatment, blinding might not be done for operators (Lines 232 to 233).

11. Initial Question: Regarding the analysis of incidence of PONV, what was used in the control group? Also, you have mentioned incidence with EAS than presenting data separately on EA and TEAS.

My Reply: Thank you for asking. The control group included in our study included: use of an inactive device, no needles with active device, no needles with inactive device, patient-controlled analgesia+ inactivated device, patient-controlled analgesia+ usual care. We have now revised this description in the text somewhat to be clearer (line 182 to 184; line 345 to 347). Our aim was to study the role of EAS, therefore, in describing the incidence of PONV, we used EAS as an overall intervention. In the fourth paragraph of the chapter, we did a subgroup analysis to determine whether EA or TEAS had treatment effect. The results showed that both EA and TEAS had significant therapeutic effects.

Reviewers’ Reply: I am still very confused about inactive device with needle. Can you clarify this a bit more? 

My Last Response: We apologize for the confusion. In the answer to Question 9, we described all of the controls. The "inactive device with needle" you mentioned here should correspond to "no needle + inactive device", which means that the needles were bent to lay flat against the skin or no needles were applied and insulated wires from the activated stimulator box with no current output were attached to the needles or the insides of arm covers[6, 7]. 

We have revised this description in the text to make this more clear (lines 361 to 369).

Reviewers’ repeat response: What is the active device without needle then?

My New Response: Thank you for your question. We are sorry for the unclear explanation. Active device without needle means no acupuncture, the activated stimulator box with a normal current output and the wire of the stimulator is pasted over the hand, without needle and stimulation. Hopefully our pictures will explain this.

13. Initial Question: What do you mean by “EAS plus other treatments”? What are the other components of the other treatments? From the data I can see the additional other treatment did not decrease the incidence, rather increased the incidence of PONV a bit compared to only EAS. Why so?

My Reply: Sorry for the undefined description. "EAS plus other treatments" means that there were other interventions in the treatment group in addition to EAS. Other treatments included the use of PCA in 6 articles and Dexameth in 1 article. Among these 7 studies, these other treatments were also applied in the control group. The variable between two groups in one article was the presence or absence of EAS. We have added an explanation in the "Incidence of PONV" section of our article.

Reviewers’ Reply: Is PCA a treatment option for managing PONV? Dexamethasone has only preventive role in PONV, it is not a therapeutic option. I am really now confused about your open unfiltered search method. 

My Last Response: Thank you for raising this question. We defined PCA and dexamethasone as “other treatments” inappropriately. PCA is not a treatment option for managing PONV. It is a pain relief method given for postoperative analgesia. PCA is routinely used for postoperative analgesia in some countries. In addition, as you stated, dexamethasone is used as routine prophylaxis, not as a treatment. Six studies explicitly emphasized the use of PCA because all six studies were also about postoperative recovery or postoperative analgesia. One study explicitly emphasized the use of dexamethasone because it focused on the efficacy of dexamethasone and TEAS. Among these 7 studies, PCA/dexamethasone was also applied in the control group. Therefore, neither PCA nor dexamethasone is a specific treatment modality for PONV and is part of the standardized treatment regimen for both groups of patients. The attached table shows the 7 studies that emphasized the usage of PCA/dexamethasone. To remove the ambiguity, we have deleted the "EAS plus other treatments" from the table and text and removed the subgroup analysis of this topic from the results. Thank you again for the helpful feedback and comments on the manuscript.

Author (year) Title of the paper.

El-Rakshy et al, 2009 Effect of intraoperative electroacupuncture on postoperative pain, analgesic requirements, nausea and sedation: a randomised controlled trial.

Gu et al, 2019 The effect of pre-treatment with transcutaneous electrical acupoint stimulation on the quality of recovery after ambulatory breast surgery: a prospective, randomised controlled trial.

Chen et al, 2015 ranscutaneous electric acupoint stimulation alleviates remifentanil-induced hyperalgesia in patients undergoing thyroidectomy: a randomized controlled trial.

Ye et al, 2008 Pain management using Han's acupoint nerve stimulator combined with patient-controlled analgesia following neurosurgery: A randomized case control study

Chen et al, 1998 The effect of location of transcutaneous electrical nerve stimulation on postoperative opioid analgesic requirement: acupoint versus nonacupoint stimulation.

Zheng et al, 2008 Effect of transcutaneous electrical acupoint stimulation on nausea and vomiting induced by patient controlled intravenous analgesia with tramadol.

Yang et al, 2015 Dexamethasone alone vs in combination with transcutaneous electrical acupoint stimulation or tropisetron for prevention of postoperative nausea and vomiting in gynaecological patients undergoing laparoscopic surgery.

Reviewers’ repeat response: Partially agree. Even though your included studies have used PCA for postoperative analgesia (possibly opioid), it should not be included as measure for PONV, rather will be a proponent of causing PONV. Yes, I agree that prophylactic dexamethasone can really have in impact on the incidence of PONV and thus should be included in the meta-analysis.

My New Response: Thank you so much for your agreement. 

17. Initial Question: Targeted and comparative literature review is missing.

My Reply: We sincerely appreciate the reviewer's insights. In the Discussion section, we have added a paragraph discussing the development of studies for meta-analysis related to our topic (line 411 to 422).

Reviewers’ Reply: Targeted literature review does not only mean history of meta-analysis related to your topic and where yours one differ with previous one. This mean analysing the previous literature related to PONV, related to your areas of PONV topic and meta-analysis, and finally where they differed, why they differed from yours meta-analysis. Also, why yours one is unique and what new you have found from them. You have to defend your findings, and negate the differing findings of others. 

My Last Response: Thank you for the specific suggestion. we rewrote the literature review and analytic discussion section. We have added the targeted and comparative literature to the discussion accordingly (lines 438 to 464).

Thank you for giving us your constructive comments. We think that the revised manuscript has become more convincing.

Reviewers’ repeat response: Again I am unable to see targeted comparative literature review. For your every outcome measures you should mention some previous literature whether it is matching or you have found a new one. If not matching, why so. Simple descriptive literature review is unnecessary here.

General discussion on PONV and its etiology are immaterial here as this is very well-known. Keep the literature review pertaining to your topic of interest.

My New Response: Many thanks for your comments and suggestions. This time, discussion has been re-organized to introduce our outcome measures from the previous literature (Lines 373 to 427). For our outcome measures, we mention some previous literature (if similar literature exists). And we agree with you that the simple descriptive literature review is unnecessary here. So, we decided to delete the section. We hope you are satisfied with our revision, and thank you again for helping us improve the quality of this paper.

Weakness:

• English language used is very lucid, has several grammatical mistakes and lacked scientific tone.

• Some design flaws are quite obvious.

• No clarity on control or sham treatment group.

• No clear comparison with already recommended pharmacological treatment group.

• No obvious risk assessment or use of scoring system.

• Several repetitions of statistical methods in the methodology and result section. Especially the result section is confusing.

• The literature review is incomplete without targeted comparative analysis.

Reviewers’ question: Why no comment here?

My Last Response: We apologize for not replying to your comments. Thank you for the reviewer’s careful reading of our manuscript. In light of all the above issues, we have carefully revised the document item by item.

Reviewers’ repeat response: Your all loopholes and gaps need to be highlighted in the limitation section clearly. 

Discussion should be started with findings of your meta-analysis than the group description. The description of groups with active or inactive with or without needles should come in the methodology or literature search section, not in the first paragraph of discussion.

My New Response: Thank you for pointing out those weakness of our manuscript. We sincerely apologize for all these shortcomings. In light of all the above issues, we have carefully revised the document item by item. For the issues (Some design flaws are quite obvious; No clear comparison with already recommended pharmacological treatment group; No obvious risk assessment or use of scoring system.) that could not be revised, we have listed them in the limitations (Lines 480 to 487).

We highly agree your comments. We moved the description of groups with active or inactive with or without needles to the literature search section. Thank you again for your advice (Lines 106 to 115).

Initial impression: Good intent of analyzing the efficacy of non-pharmacological measure in very relevant PONV which is still an intriguing issue despite abundance of antiemetics. But the manuscript still needs several modifications, refinement before I can accept it.

Reviewers’ question: Why no comment here?

My Last Response: Thank you for taking the time to read our revision and giving us your constructive comments. We have made the corresponding changes in the revised manuscript, and we think that the revised manuscript has become more rigorous and convincing. In the answer to the fifteenth question, we explained why we did not analyze the frequency of PONV. In addition, we rewrote the literature review and analytic discussion section. Thank you again for giving us the opportunity to have our manuscript reconsidered. We earnestly appreciate the reviewers’ hard work and hope that the corrections will meet with approval. Should you have any questions, please contact us without hesitation.

Reviewers’ Renewed Impression: Although the manuscript has matured much, but still there are areas which need to be improved further including the English language as well as targeted comparative literature review. I shall like to give authors another chance to improve the manuscript as it is improving in right direction. If they come up with clear explanations on the points I have raised and modify the manuscript accordingly I will give another look.

My New Response: We thank you for the positive comments and suggestions. We have tried our best to revise the manuscript according to your suggestions including grammatical mistake, mistakes in sentence organization as well as use of punctuation. As you say, we hope for help from the editor in grammar editing. Furthermore, the question about targeted comparative literature review, we rewrote the literature review and analytic discussion section. We have added the targeted and comparative literature to the discussion accordingly (Lines 373 to 427). For our outcome measures, we mention some previous literature. We hope this revision is appropriate. Thank you again for giving us the opportunity to revise our manuscript.

Reviewer #4: 

1. Please describe the treatment methods, including the treatment site, treatment schedule, electrical stimulation intensity, evaluation schedule, and the results of RCTs included in this study in more detail.

My response: Thank you for your question and suggestion. We are happy to describe the treatment methods as your suggestions. We have added a Table 2 that include the treatment site, electrical stimulation intensity, and evaluation schedule. Treatment schedule and the results of RCTs included in this study had been describe in table 1, so we have not added these two to S2 Table. Thanks again to the reviewer for professional suggestions.

2. Please provide the literature search strategy of each electronic database as a additional file in a supporting information.

My response: Thank you for your advice. We have added the literature search strategy of each electronic database as an additional file in a supporting information (S1 Table). We hope that our modification is satisfactory to you.

3. Although the quality of evidence was moderate to low, the conclusion in the abstract section is too positive. Please revise it.

My response: Thank you for pointing this out. We agree with you that the conclusion in the abstract section is too positive. We have modified the sentences in abstract section (Lines 44 to 45). Again, thanks for this point.

4. The authors provided reference 32 and 33 as the basis for the sentence “ EAS is becoming increasingly popular in clinical practice due to its repeatability and standardization of frequency, intensity and duration “ in the introduction section ( page 4, line 82-83). I wonder how this sentence relates to reference 32 and 33.

My response: We are so sorry for the mistake. “EAS is becoming increasingly popular in clinical practice due to its repeatability and standardization of frequency, intensity and duration” in the introduction section is not objective. Therefor we delete this sentence including the reference (Line 83).

Reference

1. Apfel CC, Greim CA, Haubitz I, Goepfert C, Usadel J, Sefrin P, et al. A risk score to predict the probability of postoperative vomiting in adults. Acta anaesthesiologica Scandinavica. 1998;42(5):495-501. Epub 1998/05/30. doi: 10.1111/j.1399-6576.1998.tb05157.x. PubMed PMID: 9605363.

2. Yeoh AH, Tang SS, Abdul Manap N, Wan Mat WR, Said S, Che Hassan MR, et al. Effectiveness of P6 acupoint electrical stimulation in preventing postoperativenausea and vomiting following laparoscopic surgery. Turkish journal of medical sciences. 2016;46(3):620-5. Epub 2016/08/12. doi: 10.3906/sag-1502-56. PubMed PMID: 27513234.

3. Balshem H, Helfand M, Schünemann HJ, Oxman AD, Kunz R, Brozek J, et al. GRADE guidelines: 3. Rating the quality of evidence. Journal of clinical epidemiology. 2011;64(4):401-6. Epub 2011/01/07. doi: 10.1016/j.jclinepi.2010.07.015. PubMed PMID: 21208779.

4. Alonso-Coello P, Schünemann HJ, Moberg J, Brignardello-Petersen R, Akl EA, Davoli M, et al. GRADE Evidence to Decision (EtD) frameworks: a systematic and transparent approach to making well informed healthcare choices. 1: Introduction. BMJ (Clinical research ed). 2016;353:i2016. Epub 2016/06/30. doi: 10.1136/bmj.i2016. PubMed PMID: 27353417.

5. Xin Z, Xue-Ting L, De-Ying K. GRADE in Systematic Reviews of Acupuncture for Stroke Rehabilitation: Recommendations based on High-Quality Evidence. Scientific reports. 2015;5:16582. Epub 2015/11/13. doi: 10.1038/srep16582. PubMed PMID: 26560971; PubMed Central PMCID: PMCPMC4642304.

6. Rusy LM, Hoffman GM, Weisman SJ. Electroacupuncture prophylaxis of postoperative nausea and vomiting following pediatric tonsillectomy with or without adenoidectomy. Anesthesiology. 2002;96(2):300-5. Epub 2002/01/31. doi: 10.1097/00000542-200202000-00013. PubMed PMID: 11818760.

7. Sahmeddini MA, Farbood A, Ghafaripuor S. Electro-acupuncture for pain relief after nasal septoplasty: a randomized controlled study. Journal of alternative and complementary medicine (New York, NY). 2010;16(1):53-7. Epub 2009/12/17. doi: 10.1089/acm.2009.0288. PubMed PMID: 20001536.

8. Sayed JA, MA FR, MO MA. Comparison of dexamethasone or intravenous fluids or combination of both on postoperative nausea, vomiting and pain in pediatric strabismus surgery. Journal of clinical anesthesia. 2016;34:136-42. Epub 2016/10/01. doi: 10.1016/j.jclinane.2016.03.049. PubMed PMID: 27687360.

---

## [Decision Letter · Decision Letter 3]

5 May 2023

Efficacy and safety of electrical acupoint stimulation for postoperative nausea and vomiting: A systematic review and meta-analysis

PONE-D-21-05426R3

Dear Dr. Song,

We’re pleased to inform you that your manuscript has been judged scientifically suitable for publication and will be formally accepted for publication once it meets all outstanding technical requirements.

Kind regards,

Ahmed Mohamed Maged, MD

Academic Editor

PLOS ONE

Additional Editor Comments (optional):

Reviewers' comments:

Reviewer's Responses to Questions

**Comments to the Author**

1. If the authors have adequately addressed your comments raised in a previous round of review and you feel that this manuscript is now acceptable for publication, you may indicate that here to bypass the “Comments to the Author” section, enter your conflict of interest statement in the “Confidential to Editor” section, and submit your "Accept" recommendation.

Reviewer #4: All comments have been addressed

Reviewer #5: All comments have been addressed

2. Is the manuscript technically sound, and do the data support the conclusions?

Reviewer #4: Yes

Reviewer #5: Partly

3. Has the statistical analysis been performed appropriately and rigorously? 

Reviewer #4: I Don't Know

Reviewer #5: I Don't Know

4. Have the authors made all data underlying the findings in their manuscript fully available?

Reviewer #4: Yes

Reviewer #5: Yes

5. Is the manuscript presented in an intelligible fashion and written in standard English?

Reviewer #4: Yes

Reviewer #5: Yes

6. Review Comments to the Author

Reviewer #4: Thank you for your revision. The authors have adequately addressed my comments raised in a previous round of review. this manuscript is now acceptable for publication

Reviewer #5: Hi, thank you for submitting the revision. i am going to accept this article in pleasant way. this revision will definitely make a good impression on your article.

7. PLOS authors have the option to publish the peer review history of their article (what does this mean?). If published, this will include your full peer review and any attached files.

Reviewer #4: No

Reviewer #5: No

---

## [Editor Report · Acceptance letter]

22 May 2023

PONE-D-21-05426R3 

Efficacy and safety of electrical acupoint stimulation for postoperative nausea and vomiting: A systematic review and meta-analysis 

Dear Dr. Song:

I'm pleased to inform you that your manuscript has been deemed suitable for publication in PLOS ONE. Congratulations! Your manuscript is now with our production department. 

Kind regards, 

on behalf of

Professor Ahmed Mohamed Maged 

Academic Editor

PLOS ONE